# The ion trap aerosol mass spectrometer: field intercomparison with the ToF-AMS and the capability of differentiating organic compound classes via MS-MS

Johannes R. W. Fachinger[1], Stéphane J. Gallavardin[1,2], Frank Helleis[3], Friederike Fachinger[1], Frank Drewnick[1], Stephan Borrmann[1,2]

[1]Particle Chemistry Department, Max Planck Institute for Chemistry, Mainz, 55128, Germany
[2]Institute for Atmospheric Physics, Johannes Gutenberg University Mainz, Mainz, 55128, Germany
[3]Max Planck Institute for Chemistry, Mainz, 55128, Germany

*Correspondence to*: Stephan Borrmann (stephan.borrmann@mpic.de)

**Abstract.** Further development and optimisation of a previously described ion trap aerosol mass spectrometer (IT-AMS) are presented, which resulted in more reproducible and robust operation and allowed for the instrument's first field deployment. Results from this 11-day long measurement indicate that the instrument is capable of providing quantitative information on organics, nitrate, and sulphate mass concentrations with reasonable detection limits ($0.5 - 1.4$ µg m$^{-3}$ for 1 h averages), and that results obtained with the IT-AMS can directly be related to those from Aerodyne aerosol mass spectrometers. The capability of the IT-AMS to elucidate the structure of fragment ions is demonstrated via an MS$^4$ study on tryptophan. Detection limits are demonstrated to be sufficiently low to allow for MS$^2$ studies not only in laboratory, but also in field measurements under favourable conditions or with the use of an aerosol concentrator. In laboratory studies the capability of the IT-AMS to differentiate $[C_4H_y]^+$ and $[C_3H_yO]^+$ fragments at the nominal *m/z* 55 and 57 via their characteristic fragmentation patterns in MS$^2$ experiments is demonstrated. Furthermore, with the IT-AMS it is possible to distinguish between fragments of the same elemental composition ($[C_2H_4O_2]^+$ at *m/z* 60 and $[C_3H_5O_2]^+$ at *m/z* 73) originating from different compound classes (carboxylic acids and sugars) due to their different molecular structure. These findings constitute a proof of concept and could provide a new means of distinguishing between these two compound classes in ambient organic aerosol.

**Keywords:** ion trap; aerosol mass spectrometry; sugar; carboxylic acid; structure determination

## 1 Introduction

Despite the fact that atmospheric aerosol particles have an important influence on air quality, public health, and the climate system, the knowledge on the influence of individual aerosol particle chemical components remains limited (Fuzzi et al., 2015). One main contributor to fine particulate matter is organic aerosol (Kanakidou et al., 2005), which includes a large variety of different, to a large part unknown, organic molecules (Goldstein and Galbally, 2007).

Much work has been invested in the past in order to chemically characterise the organic material present within the ambient aerosol, mostly using mass spectrometric techniques (Hoffmann et al., 2011). For off-line analyses of filter samples from atmospheric aerosol particles, mass spectrometry is often coupled to chromatographic methods for prior separation of the organic compounds (Pratt and Prather, 2012), while with the most commonly used on-line techniques either single particles (Murphy, 2007; Silva and Prather, 2000) or small ensembles of particles (Canagaratna et al., 2007) are analysed without prior separation of substances.

Currently, the most widely used instrument deploying the latter method is the Aerodyne aerosol mass spectrometer (AMS, Aerodyne Inc.; Jayne et al., 2000), in which ensembles of particles are flash-vaporised at typically ~600 °C and the evolving vapour is ionised by electron impact ionisation before mass spectrometric analysis, usually with a time of flight mass spectrometer (ToF-AMS; Drewnick et al., 2005; DeCarlo et al., 2006). Since a large number of different molecules are analysed simultaneously, a mathematical deconvolution algorithm has to be applied to the acquired mass spectra in order to obtain information on different particle components (Allan et al., 2004). Due to the thermal desorption (which already might cause some fragmentation) and additional "hard" electron impact ionisation, molecules are highly fragmented, which means a partial loss of the original molecular information (e.g., typically no molecular ion is observed). On the other hand, since different molecules containing the same sub-structure will be reduced to the same fragment ions, this considerably reduces the complexity of information when dealing with mixtures of a large variety of different organic compounds (Hoffmann et al., 2011) while some important information on the original molecular structures are still retained in these common fragment ions, which enables the determination of "types" of organic particle constituents using positive matrix factorisation (Zhang et al., 2011).

The differentiation of organic components can be further constrained and improved using the higher resolution ToF-AMS, which has a mass resolution of ~2000 or ~4000 in two modes providing higher and lower sensitivity, respectively (DeCarlo et al., 2006). With these resolutions, it is possible to distinguish between isobars, i.e., ion fragments of the same nominal mass to charge ratio ($m/z$), but with different elemental compositions. For example, a prominent mass spectral signal at $m/z$ 55 is observed in both hydrocarbon-like (traffic-related) and cooking-related organic aerosol. While this signal seems to be related mostly to $[C_4H_7]^+$ in hydrocarbon-like organic aerosol, an additional large fraction of $[C_3H_3O]^+$ at the same nominal $m/z$ is found for cooking-related organic aerosol (Sun et al., 2011).

While information on elemental composition of the fragment ions can be obtained with the ToF-AMS, it does not allow for the differentiation between fragment ions of the same elemental composition, but with different structural formulas (i.e., isomeric ions). Such information can be obtained using ion trap mass spectrometers (March, 1997), which allow not only for measuring a "classic" mass spectrum (MS), but enable also $MS^n$ studies. In such experiments, ions of a single nominal $m/z$ are first isolated and then fragmented by collisions with buffer gas atoms (collision induced dissociation, CID). The resulting fragment ions can be either mass scanned, which provides the mass spectrum of the fragment ions (MS-MS, or $MS^2$), or again ions of a certain $m/z$ can be separated and fragmented, and so on ($MS^n$). From the fragment ions and fragmentation pathways, conclusions on the molecular structure of the parent ion can be drawn (March, 1997; McLafferty and Tureček,

1993). Additionally, ion/molecule reactions inside the ion trap can be utilized in order to differentiate between isobaric or isomeric ions (e.g., Kascheres and Cooks, 1988).

Due to their capability to elucidate the molecular structure of organic molecules, ion traps are often coupled to "soft" ionisation techniques, in order to preserve the molecular structure of the compounds of interest as much as possible. For atmospheric applications, e.g., vacuum ultraviolet single photon ionisation (Hanna et al., 2009; Schramm et al., 2009), proton transfer reaction (Thornberry et al., 2009), and atmospheric pressure chemical ionisation (Vogel et al., 2013) have been successfully applied in conjunction with ion trap mass spectrometry. Kürten et al. (2007) introduced an instrument which represents a synthesis of an AMS ionisation chamber (thermal desorption / electron impact ionisation) with a quadrupole ion trap mass spectrometer; a similar instrument was also developed by Harris et al. (2007). With these systems, a strong reduction in complexity of ambient organic aerosol mass spectra (which consist of a large number of different organic molecules) is achieved compared to "soft" ionisation techniques, while still some additional information, e.g., on molecular functionality can be obtained which is not accessible from ToF-AMS measurements.

Here, we present further design improvements of the ion trap aerosol mass spectrometer (IT-AMS) originally introduced by Kürten et al. (2007). Technical improvements enable more robust and reproducible measurements with this instrument since the work of Kürten et al. (2007) and allowed for the instrument's first field deployment. We show how the IT-AMS compares with a regular ToF-AMS and demonstrate its $MS^n$ capability up to $MS^4$, also providing an estimate of the related detection limits. Finally, from $MS^2$ studies on various compounds, we demonstrate how the IT-AMS is capable of distinguishing between isomeric fragment ions, which could provide a means to distinguish between carboxylic acids and sugars in organic aerosol, and with more extensive calibration potentially also to quantify their relative fractions in more complex mixtures. Since sugars and carboxylic acids can be associated with different aerosol sources (sugars originate e.g. from biomass burning or primary biological material, while carboxylic acids originate e.g. from photo-oxidation of organic precursors (Graham et al., 2003)), this would help in further improving the differentiation of various organic aerosol components and therefore in source apportionment of atmospheric organic aerosol.

## 2 Instrumental developments

The original setup and the working principle of the ion trap aerosol mass spectrometer (IT-AMS) are described in detail in (Kürten et al., 2007). Here, only a brief overview of the instrument is given (Fig. 1). The instrument consists of the same commercially available particle inlet and ionisation chamber as contained in the AMS (Jayne et al., 2000), coupled to a home-built quadrupole ion trap (Kürten et al., 2007). Sub-micron sized particles entering the instrument are focused by a Liu-type aerodynamic lens (Liu et al., 1995). A shutter located behind the inlet can be either closed to block the particle beam (measurement of instrumental background) or opened (particle beam plus background), enabling the calculation of "difference" mass spectra (open minus closed) originating purely from the particle beam (Canagaratna et al., 2007). In the ionisation chamber the particles are flash-vaporised on a hot tungsten surface (~600 °C), and the resulting vapour is ionised

by electron impact ionisation (70 eV). The ions are extracted through ion optics to the quadrupole ion trap, which consists of a ring electrode and two hyperbolical end cap electrodes with modified angle geometry with $r_0 = 1$ cm and $z_0 = r_0/\sqrt{1.9} = 0.725$ cm, where $r_0$ and $z_0$ are the shortest distances from the centre of the trap to the ring and end cap electrodes, respectively. In the ion trap, the ions can be stored and manipulated; ions leaving the trap are then detected using a channeltron (KBL510, Sjuts). High purity helium (6.0, Westfalen AG) is used as buffer gas in the ion trap. Mass spectral resolution depends on the settings (Kürten et al., 2007) and was ~400 for the measurements described here.

While the instrument described in (Kürten et al., 2007) was capable of providing quantitative information on aerosol components, it suffered from a lack of user-friendliness and from limited long-term stability and reproducibility of the settings. Therefore, in order to prepare the instrument for field measurements and regular laboratory applications, several hardware (electronical and mechanical) as well as instrument control software modifications were performed, all aiming at a more robust, versatile, and user-friendly operation of the instrument. The operation principles for the MS and MS$^n$ measurements, e.g., mass range extension, ion isolation, and ion excitation and fragmentation, were nevertheless kept the same. We only describe the most important changes in detail here; these as well as some other, minor modifications are also described in (Fachinger, 2012).

Most importantly, three hardware modifications related to the mechanical design of the instrument were realised (see Fig. 1, inserts):

-   In the ion trap, originally ceramic washers of 2.87 mm ± 25 µm thickness were used by Kürten et al. (2007) as spacers between the ring and end cap electrode, and the electrodes were held by four threaded bars insulated by ceramic shells with a play of 0.5 mm each. Due to these rather large allowances for tolerance, the electrodes could not be assembled reproducibly enough to maintain the geometry (i.e., without rotation or tilting of the electrodes), and after each re-assembly of the electrodes the voltage settings for the various operations (e.g., trapping, scanning, resonant excitation) had to be re-tuned. To avoid this, now four ruby spheres (diameter 6 mm ± 0.635 µm) sitting in precise countersinks (1.3 mm ± 10 µm) are used instead of the ceramic washers for a more defined mounting (insert B). This allows for a more reproducible assembly of the electrodes (i.e., invariant geometry), and consequently more reproducible voltage settings without needing to re-tune after each re-assembly of the ion trap electrodes.

-   In order to ensure stable and reproducible ion signal intensities, constant helium pressure in the ion trap is needed, which is achieved in the IT-AMS by a constant inlet mass flow of helium and constant pumping speed. In the original setup, this helium inlet flow was regulated via a needle valve, and therefore changed with changing ambient (upstream) pressure (over the course of four days, a relative standard deviation of 2 % was found at an average pressure of $2 \cdot 10^{-5}$ hPa measured outside the trap). Now, constant inlet mass flow is provided by a critical orifice (30 µm inner diameter) with constant upstream pressure ensured by a pressure-controlled mass flow controller (Bronkhorst High-Tech B.V., EL-PRESS P-502-C and F-004AC with a specified flow rate of ≤0.7 L min$^{-1}$; insert A). With this system, the pressure inside the system ($7 \cdot 10^{-5}$ hPa, measured outside the trap) was found to be stable within less than 1 % (relative standard deviation of 15 s data) over the course of eleven days.

- The ion trap needs a pulsed ion source so that ions are only generated and transmitted to the ion trap during the trapping phase, but not during the analysis phase. This was realised in the original setup by gating of the ion source cage voltages (Kürten et al., 2007): only during the trapping phase, electrons were accelerated into the ion source cage, while in the other phases, they were deflected from it. This led to an instable filament emission current directly after switching the voltages, potentially due to the build-up of space charges. Now, a more stable filament emission current is achieved by the use of a modified filament which allows for pulsed ion source operation: In the original filament, the emission of the electrons (defined by the filament current) and the voltage of the filament's deflection plate were electronically coupled in such a way that emitted electrons always were repelled by the deflection plate and accelerated away from the filament and towards the ion cage. In the modified filament the deflection plate is electronically decoupled from the filament, such that electrons are emitted continuously, but the voltage of the deflection plate can be set independently and switched from negative to positive sign. By this means, electrons emitted by the filament are now either deflected or absorbed by this deflection plate (insert C), depending on whether they are needed in the ion source or not. This controlled absorption of the electrons (instead of only repelling them from the ion cage) allows for a more defined gating of the electrons and avoids the potential build-up of space charges.

The IT-AMS is controlled via a program written in LabVIEW (v.8.5, National Instruments), which is also utilised for data acquisition. The original software was very rudimentary and did not contain several important features needed for a routine deployment of the instrument. Therefore this software was extended and now includes the option for a semi-automatic tuning of operation parameters, i.e., the instrument is programmed by a user-adaptable text file to automatically scan the various (five for MS, nine for $MS^2$) parameters of interest and to save the results, which then can be inspected to find the optimal set of tuning parameters. The software now also allows for much more flexibility in the measurement types and their operating conditions (MS, $MS^2$, $MS^{n>2}$ ($n \leq 5$), mass range extension), programming long series of measurements, and the control of the shutter to enable automatic switching between open / closed measurements, as described above. Furthermore, all instrumental settings and parameters are now saved along with the mass spectra after each measurement cycle.

## 3 Laboratory and field measurements

In the laboratory studies performed separately with the IT-AMS and the ToF-AMS, respectively, particles were typically generated from an aqueous solution of the respective substance (Table 1) using a nebuliser (model 3076, TSI Inc.). Oleic acid was dissolved in ethanol instead of water (in case of the ToF-AMS, an aqueous suspension was used), while butyl valerate in both cases was leaked as vapour directly into the instrument. After generation, the aerosol was dried using two consecutive silica gel diffusion driers before sampling with the IT-AMS or high resolution ToF-AMS (DeCarlo et al., 2006), respectively. For the determination of the $MS^n$ detection limits using tryptophan (Sect. 4.2), monodisperse aerosol (130 nm

mobility diameter) was generated by classifying the dried particles using a differential mobility analyser (model 3081, TSI Inc.). Parallel measurement of particle number concentration using a condensation particle counter (model 3025A, TSI Inc.) enabled the calculation of the sampled aerosol mass concentration, assuming spherical particles with the bulk density of tryptophan (1.34 g cm$^{-3}$).

Continuous measurements of ambient aerosol using the IT-AMS and a high resolution ToF-AMS were concurrently performed on the Mt. Kleiner Feldberg (Central Germany) from 29 August to 09 September 2011, within the context of a larger measurement campaign (Sobanski et al., 2016). Both instruments were sampling in parallel through two separate inlets (7 m a.g.l), which were located in about 5 m distance from each other. Since no local sources were close to the measurement site, only regional background aerosol was measured, which can be expected to be homogeneously distributed on this spatial

scale. The IT-AMS was measuring with a time resolution of 30 s (10 s particle beam blocked, 10 s open, with 5 s waiting time between each half-cycle; during each 10 s interval 100 mass spectra were averaged, each acquired after 50 ms ion accumulation time), the ToF-AMS with a resolution of 60 s (containing 15 s beam blocked, 15 s open, and 30 s size distribution measurement).

In all laboratory and field measurements, ions were trapped over 50 ms at a radiofrequency (1.3 MHz) drive voltage of 250

to 700 V amplitude (zero-to-peak) applied to the ion trap ring electrode. To reach the maximum $m/z$ of 135 used within this work, in the laboratory experiments – if needed – the mass range was extended by applying an additional voltage of 400 mV amplitude (zero-to-peak) with a frequency of 400 kHz to both end cap electrodes. In MS$^{n}$ experiments, ions of $m/z$ of interest were isolated (typically within a range of ± 5 $m/z$, but sometimes up to ± ~15 $m/z$) by broad band excitation using a filtered noise field (Julian and Cooks, 1993) before they were fragmented using collision induced dissociation (CID, see parameters

in Table 2).

Measured IT-AMS mass spectral signals were converted to ion rates (number of measured ions divided by the length of the trapping phase) and integrated to unit mass resolution (UMR) mass spectra using home-built procedures in IGOR Pro (v.6.22, WaveMetrics Inc.) and MATLAB (R2006, MathWorks). High resolution ToF-AMS data were analysed using SQUIRREL v.1.44 and PIKA v.1.04 and higher (SQUIRREL, 2016).

From the ToF-AMS field data, organics, nitrate, and sulphate mass concentrations were determined using the fragmentation pattern table (Allan et al., 2004) within SQUIRREL (v.1.51H), which was adjusted to correct for background effects using routinely performed measurements of particle-free air (obtained by inserting a high efficiency particulate arrestance filter in the sampling line). Ionisation efficiency of the ToF-AMS was determined prior to the campaign applying the established method (Canagaratna et al., 2007) using dried NH$_4$NO$_3$ particles of known mobility diameter (400 nm). A collection

efficiency of 50 % was applied following Canagaratna et al. (2007), which resulted in good agreement with other co-located measurements (Fachinger, 2012): Comparison of the 1 min averaged time series of PM$_1$ calculated by summing all ToF-AMS measured species plus independently measured black carbon (using a Multi-Angle Absorption Photometer MAAP, model 5012, Thermo Scientific) with measurements of total PM$_1$ (using an Environmental Dust Monitor EDM 180, Grimm

Aerosol Technik GmbH & Co. KG) gave a correlation of Pearson's $R^2 = 0.91$ and a slope of 1.11, i.e. good agreement within the uncertainty of the ToF-AMS of ~30 %.

For the IT-AMS field data, difference mass spectra were calculated from the UMR mass spectra obtained during beam open and blocked time. From these, species-resolved total ion rates for organics, nitrate, and sulphate were calculated using a

simplified fragmentation pattern table (due to reduced signal to noise ratio) by summing the mass spectral signal of the $m/z$ containing the respective species' dominant ions in the $m/z$ range 30 – 106 (nitrate: $m/z$ 30, 46; sulphate: $m/z$ 48, 64, 80, 81, 98; organics: all other $m/z$ except 32, 39, 40).

## 4 Results and discussion

### 4.1 Measurement of ambient aerosol: comparability to ToF-AMS results

Figure 2 shows a comparison of the average mass spectra from the IT-AMS and the ToF-AMS, acquired during 11 days of field measurement. The $m/z$ 30 and above are colour-coded for the species (organics, nitrate, sulphate) with which they are dominantly associated (see Sect. 3). Here, we discuss general trends on the basis of the campaign average in order to minimise the statistical uncertainty. The same features are typically also visible when comparing 1 h averaged mass spectra; in that case the observed trends are more distinct at higher absolute mass concentration (i.e., at higher signal to noise ratio).

Strong differences between the mass spectra are observed in the $m/z$ range below 30. Apart from the potential influence of lower ion transmission or lower trapping efficiencies for low $m/z$ ions in the IT-AMS, this is probably mostly due to the strong influence of charge-transfer reactions in the ion trap during the trapping phase in this $m/z$ range, e.g., involving $N^+$, $O^+$, $N_2^+$, or $O_2^+$ (Dotan et al., 1997; Hierl et al., 1997). $m/z$ 28 ($N_2^+$), 32 ($O_2^+$) and 40 ($Ar^+$), but also $m/z$ 44 ($CO_2^+$) are depleted in the IT-AMS compared to the ToF-AMS mass spectrum, likely due to formation of more stable ions by charge-

transfer reactions in the ion trap (Ottens et al., 2005). Furthermore, the ratio of $m/z$ 18 to $m/z$ 19 ($H_2O^+$ to $H_3O^+$) is much smaller (by more than 99 %) in the IT-AMS mass spectrum compared to that of the ToF-AMS, probably caused by proton transfer reactions occurring in the ion trap ($H_2O + H_2O^+ \rightarrow H_3O^+ + \cdot OH$, (Cole et al., 2003)). In laboratory experiments, we found that with increasing ion accumulation and reaction times, this ratio decreases exponentially, converging to a ratio of ~1:1 at accumulation times ≥200 ms. Beside these differences, a plateau of more or less constant relative signal intensity is

observed from $m/z$ 20 to 27 for the IT-AMS (Figure 2, upper panel), but not for the ToF-AMS. Kürten (2007) found that this plateau disappears (i.e., the mass spectral pattern in this $m/z$ range becomes more similar to that of the ToF-AMS) when an additional reaction time is applied to allow for collisional cooling (Wu and Brodbelt, 1992). It can therefore be considered an artefact. In this work, we focus on $m/z$ 30 and above, since the deconvolution of ion signal below $m/z$ 30 is complicated due to the contribution of several species to most single $m/z$ (Allan et al., 2004) and the relatively strong influence of artefacts in

this $m/z$ range.

Considering the organics signal at $m/z$ >30, most strikingly, an increasing signal ratio (IT-AMS / ToF-AMS) with increasing $m/z$ is observed (Figure 2, lower panel). This probably is explained by different ion transmission efficiency curves with

respect to *m/z* for the two instruments, which can easily be accounted for by calibrations. Importantly, though, the fragmentation pattern (Figure 2, upper panel) is comparable between both instruments (Pearson's $R^2$ of 0.78; if *m/z* 44 – which is influenced by charge-transfer reactions inside the trap, see above – is disregarded, $R^2 = 0.90$), despite the much longer residence time of the ions in the ion trap (50 ms accumulation time) compared to the ToF-AMS. Thus results obtained

with the IT-AMS, including MS-MS measurements, are directly transferable to ToF-AMS measurements. This comparability needs to be validated also for other accumulation and reaction times.

In contrast, signals of *m/z* mostly related to sulphate fragments are strongly depleted in the IT-AMS compared to the ToF-AMS mass spectrum, and also the fragmentation pattern is different. The signal ratio *m/z* 48 to *m/z* 64, which are dominated by the sulphate ion fragments $SO^+$ and $SO_2^+$, is 0.8 in the ToF-AMS mass spectrum, but 1.7 in that of the IT-AMS, despite

the opposing trend of ion transmission efficiency. Both decomposition of the ions and charge-transfer reactions (with different efficiency for different ions) could cause this ion depletion and fragmentation pattern changes. This needs to be further investigated, also as a function of accumulation and reaction times. Contrary, for nitrate-related ions, no depletion in the IT-AMS has been found.

Figure 3 shows the measured organics, nitrate, and sulphate concentration time series for both instruments. For the IT-AMS,

the results are reported both in ion rates and in mass concentrations, which are calculated using the linear relationships with the ToF-AMS (Fig. 3, inlays). For all three species, the time series measured with IT-AMS and ToF-AMS correlate linearly over the range of observed mass concentrations (Fig. 3), confirming the capability of the IT-AMS to quantitatively measure species mass concentrations independently using adequate calibrations for all species, as demonstrated for nitrate by Kürten et al. (2007). The squared Pearson's correlation coefficients for 10 min (1 h) averaged time series are 0.82 (0.88) for

organics, 0.68 (0.65) for nitrate, and 0.37 (0.58) for sulphate (Fig. 3). The much lower correlation coefficient for sulphate compared to the other species is in agreement with the observation of much lower response (and therefore, lower ion rates and signal-to-noise ratio) of the IT-AMS for sulphate than for nitrate and organics (Figs. 2 and 3). Also the fact that the observed range of mass concentrations for sulphate was smaller than for organics and nitrate might have added to the less tight correlation for this species.

From the calibration of the IT-AMS against the ToF-AMS, furthermore relative ionisation efficiency (RIE) values for sulphate and organics can be calculated for the IT-AMS. The relative ionisation efficiency is a constant factor with which the ionisation efficiency (determined in calibrations using ammonium nitrate) is multiplied in order to get the species-dependent ionisation efficiency. In order to determine these RIEs, the slope obtained for nitrate from the correlation depicted in Fig. 3 (inlay) is related to those determined for organics and sulphate, respectively. By this means, RIE values of 0.4 for sulphate

and of 1.7 for organics were found. For organics, this is slightly higher than the RIE value of 1.4 used for the ToF-AMS, consistent with the slightly higher ion transmission of the IT-AMS for larger *m/z* (to which mostly organics are contributing). For sulphate, the RIE of 0.4 is much smaller than the sulphate RIE value typically used for the ToF-AMS (1.2), consistent with the depletion of sulphate-related ions in the IT-AMS, as described above. It also has to be kept in mind that even

without those influences, not exactly the same RIE values as for the ToF-AMS can be expected due to the use of a simplified fragmentation pattern table (see Sect. 3).

By calibrating the IT-AMS for nitrate as demonstrated by Kürten et al. (2007), with the RIE values determined here the mass concentrations of sulphate and organics can be directly obtained from IT-AMS measurements. Note, however, that RIE values might change with different accumulation and reaction times, and therefore need to be newly measured when changing the instrumental settings. The uncertainty of the derived mass concentrations can be estimated to 30 % (which includes the uncertainty due to ionisation efficiency, relative ionisation efficiency, and collection efficiency), the same as usually estimated for ToF-AMS measurements. Note that with the IT-AMS, unlike the ToF-AMS, ammonium mass concentration cannot be determined due to artefacts in the $m/z$ range <$m/z$ 30, as described above. Another species typically reported from ToF-AMS measurements, non-refractory chloride, in principle should be possible to detect with the IT-AMS (dominant mass spectral lines at $m/z$ 35 and 36), but has not been observed during this measurement due to very low mass concentrations (campaign average of 0.04 µg m$^{-3}$ found with the ToF-AMS).

The mass concentration time series derived from the IT-AMS measurements using the linear correlation with the ToF-AMS measurement (Fig. 3, inlays) within their uncertainty agree well with those of co-located measurements: the sum of black carbon (from MAAP) with IT-AMS sulphate, nitrate, and organics and corrected for the missing species ammonium by assuming fully neutralised aerosol (as expected for regional background aerosol and validated by the ToF-AMS measurements) correlates well with the total PM$_1$ mass concentration measured with the EDM (slope = 1.03, R$^2$ = 0.64 for 1 h data).

Detection limits for organics, nitrate, and sulphate are calculated following the method by Drewnick et al. (2009). Detection limits for 10 min averages are $(3.7 \pm 1.1)$ µg m$^{-3}$ for organics, $(1.3 \pm 0.4)$ µg m$^{-3}$ for nitrate, and $(1.3 \pm 0.4)$ µg m$^{-3}$ for sulphate (for 1 h averages: $(1.4 \pm 0.4)$ µg m$^{-3}$ for organics, $(0.5 \pm 0.2)$ µg m$^{-3}$ for nitrate, $(0.7 \pm 0.2)$ µg m$^{-3}$ for sulphate). The higher detection limit for organics can be explained by the fact that the mass spectral signal is distributed over a much larger number of $m/z$ (Fig. 2), thereby lowering the signal-to-noise ratio. For all three species, the detection limits found for the IT-AMS are two to three orders of magnitude higher than those reported for the ToF-AMS (DeCarlo et al., 2006; Drewnick et al., 2009), due to much lower ion rates (by three orders of magnitude). Thus, the IT-AMS is capable of providing absolute mass concentrations of organics, nitrate, and sulphate at typical ambient concentrations, e.g., in urban areas (i.e., in the order of several µg m$^{-3}$) after application of adequate calibrations.

## 4.2 Prospects and limitations of MS$^n$ studies

While the IT-AMS has lower sensitivity compared to the ToF-AMS, it allows for more detailed, in-depth studies of the measured aerosol. Using MS$^n$ studies, it is possible to derive structural information on the measured ions, while with the ToF-AMS, only information on their elemental composition can be obtained.

In this section, we discuss the potential of MS$^n$ studies with the IT-AMS exemplarily by means of results obtained for two compounds: glutathione and tryptophan (Table 1). Their "classic" mass spectra, obtained with the IT-AMS, are shown in

Figs. 4a (glutathione) and 5a (tryptophan). Both show prominent ion signals at $m/z$ 130, which judging from the structure of the respective parent molecules are likely from $[C_5H_8O_3N]^+$ (glutathione) and $[C_9H_8N]^+$ (tryptophan). With the ToF-AMS, these would be possible to distinguish only in the mode of highest mass resolution (~4000), but not in the more sensitive mode of lower mass resolution (~2000) (DeCarlo et al., 2006). With the IT-AMS, they can be distinguished by means of

MS-MS ($MS^2$): CID of $m/z$ 130 yields the fragment ions $m/z$ 103 ($[C_8H_7]^+$) and $m/z$ 128 ($[C_9H_6N]^+$) for tryptophan (Fig. 5b), but $m/z$ 83 ($[C_4H_5ON]^+$) and $m/z$ 84 ($[C_4H_6ON]^+$) for glutathione (Fig. 4c), revealing the different nature of their respective parent ions. It has to be noted that CID in these experiments is found to potentially affect ions in a range of $\pm$ 1 around the $m/z$ of interest, i.e., in the case of glutathione at least parts of the signal at $m/z$ 83 could also originate from CID of $m/z$ 129. The resulting ions (from $MS^2$) can be further fragmented in order to obtain more detailed information on their molecular

structure, as shown for tryptophan in Fig. 5c,d: CID of $m/z$ 103 yields $m/z$ 77 ($[C_6H_5]^+$) ($MS^3$); the thereby formed ion at $m/z$ 77 can be further fragmented to $m/z$ 50 ($[C_4H_2]^+$) and $m/z$ 51 ($[C_4H_3]^+$) ($MS^4$). When trying to fragment these ions ($MS^5$), no ions (neither remaining parent ions at $m/z$ 50 / 51, nor any fragments) could be detected. This could be because the ions are very stable and therefore are removed from the ion trap, even at very low amplitudes for resonant excitation; or because they are fragmented to ions of $m/z$ <20, which were not analysed in the present study.

Further information on the nature of the ions observed in the "classic" MS can be obtained by comparing the CID products of different ions from the original mass spectrum (i.e., $MS^2$ of these ions) with those of the fragmentation chain ($MS^{3-5}$) of larger (parent) fragment ions: While $MS^2$ of $m/z$ 103, 77, and 50 results in the same fragment ions as observed in the respective $MS^{3-5}$ studies of $m/z$ 130 presented above, $MS^2$ of $m/z$ 51 yields different results. Here, $m/z$ 39 ($[C_3H_3]^+$) and $m/z$ 63 ($[C_5H_3]^+$) are formed while in the fragmentation chain no further fragmentation was observed for this ion. Therefore,

it can be concluded that additionally to the fragmentation product of $m/z$ 77 (i.e., $[C_4H_3]^+$), also the doubly charged ion $[C_8H_6]^{2+}$ contributes to $m/z$ 51 in the "classic" MS of tryptophan to at least ~10 %. The ions $[C_4H_3]^+$ and $[C_8H_6]^{2+}$ would not be possible to distinguish with the ToF-AMS.

The detection limits of the tryptophan $MS^n$ experiments were calculated following the method by Drewnick et al. (2009) from the standard deviations of the signal intensities at the investigated $m/z$ in a blank $MS^n$ measurement, and the signal

intensity enhancements at the same $m/z$ during $MS^n$ measurements of known tryptophan mass concentrations. All detection limits were calculated for averages of 3000 mass spectra, which correspond to measurement times of 6, 16, 28, and 46 minutes for the full MS, $MS^2$, $MS^3$, and $MS^4$ cycles, respectively. The corresponding detection limits were found to be ~0.6, 7, 13, and 30 µg m$^{-3}$ at the given measurement conditions. This suggests that $MS^2$ studies might be feasible under favourable conditions at ambient concentrations, while $MS^{n>2}$ studies are only possible at high mass concentrations, as, e.g., in smog

chamber studies or by applying an aerosol concentrator.

**4.3 Differentiation of organic compound classes**

In this section, we investigate how the IT-AMS is capable of distinguishing fragments of the same nominal mass, but with different molecular and/or structural formulas. This makes it not only possible to distinguish, e.g., between hydrocarbon-like

and oxygenated organic species (Zhang et al., 2011), but potentially also between species with different functional groups, which would allow for the assignment of measured organic species to different compound classes. This potential is investigated by means of MS$^2$ studies on various substances from different compound classes, particularly sugars and carboxylic acids (Table 1).

**4.3.1 Differentiation of ions of the same nominal mass, but with different molecular formulas**

The differentiation of ions of the same nominal $m/z$, but with different molecular formulas allows, e.g., for the distinction between hydrocarbon-like and oxygenated organic species. This kind of information is also accessible with a high resolution ToF-AMS, given the resolution at the respective $m/z$ is sufficiently high, and is commonly retrieved from such data (Zhang et al., 2011). Here, we demonstrate how the IT-AMS is also capable of providing such information. Figure 6 shows a
compilation of results from MS$^2$ studies on $m/z$ 55 and 57 for a variety of compounds. Also shown is the relative contribution of the major ions at these $m/z$ measured with a ToF-AMS (left panels): $[C_3H_3O]^+$ and $[C_4H_7]^+$ at $m/z$ 55, and $[C_3H_5O]^+$ and $[C_4H_9]^+$ at $m/z$ 57.

When fragmenting the ions at $m/z$ 55 and 57 in the IT-AMS (MS$^2$), clear differences in the resulting fragmentation patterns are observed between the different compounds. For sugars (glucose, saccharose, mannitol, levoglucosan; reddish/brownish
colours in Fig. 6), but also for succinic and glutaric acid (carboxylic acids, bluish colours in Fig. 6), fragmentation occurs predominantly by loss of CO (28 Da, resulting in $m/z$ 27 and 29 ions from parent ions at $m/z$ 55 and 57, respectively). This points to $[C_3H_yO]^+$ as the dominant parent ion for these compounds. Additionally, loss of $CH_2O$ (30 Da) is observed for MS$^2$ of $m/z$ 57, yielding the fragment ion $m/z$ 27 ($[C_2H_3]^+$), while for MS$^2$ of $m/z$ 55 for some of these compounds, loss of $C_2H_2$ (26 Da) results in the fragment ion $m/z$ 29 ($[COH]^+$). Ion recovery (i.e., the total signal of all fragment ions detected in MS$^2$
divided by the concurrent loss in signal of the parent ion) is rather low for these compounds for MS$^2$ of $m/z$ 55 (<20 %), and slightly higher for $m/z$ 57 (~40 %).

In contrast, MS$^2$ studies on oleic acid and butyl valerate reveal the dominant presence of $[C_4H_y]^+$ fragment ions at the nominal masses $m/z$ 55 and 57, consistent with the structure of the respective molecules (with long hydrocarbon chains, greenish colours in Fig. 6) and with results from corresponding ToF-AMS measurements (Fig. 6, left): For $m/z$ 55,
predominantly loss of $H_2$ (2 Da), $CH_4$ (16 Da), and $C_2H_2$ (26 Da) is observed. Less abundant is the fragment ion $m/z$ 27, originating from the loss of $C_2H_4$ (28 Da). For $m/z$ 57, predominantly fragmentation to $m/z$ 41 (loss of $CH_4$, 16 Da) is found. Ion recovery for MS$^2$ on these $[C_4H_y]^+$ fragments was found to be much better (50 – 100 %) than for the $[C_3H_yO]^+$ fragments discussed above.

For pinonic acid and PEG200 (polyethylene glycol; see Table 1), no clear picture is obtained. For both compounds, MS$^2$
studies suggest dominance of $[C_4H_7]^+$ at $m/z$ 55, with fragmentation patterns similar to those of oleic acid and butyl valerate. While this is comprehensible from the molecular structure for pinonic acid, for which also ToF-AMS results show a strong fraction of $[C_4H_7]^+$ at $m/z$ 55, for PEG200, dominance of $[C_3H_3O]^+$ would have been expected both from the molecular structure and from the ToF-AMS measurement (Fig. 6). This needs to be further investigated; it might be possible that the

oligomeric structure of PEG200 plays a role here, which could lead to multiply charged larger ion fragments, and to ion rearrangements, and therefore to a different molecular structure of the ion with the elemental composition $[C_3H_3O]^+$ and a different $MS^2$ mass spectrum. For $m/z$ 57, on the other hand, both PEG200 and pinonic acid show fragmentation patterns originating from $[C_3H_5O]^+$, and only small contributions at $m/z$ 41 which likely originate from fragmentation of $[C_4H_9]^+$. This

is consistent with the absence of straight hydrocarbon chains in the molecular structures of these compounds (Table 1), and with the results from the respective ToF-AMS measurements (Fig. 6, left).

These findings show that it is generally possible to distinguish between compounds containing straight hydrocarbon chains and such that do not contain them by $MS^2$ experiments on $m/z$ 55 and 57. Our results suggest that combined information from the $MS^2$ fragmentation patterns and the corresponding ion recoveries could facilitate the quantitative determination of

the relative contributions of the different ions ($[C_3H_yO]^+$ vs. $[C_4H_y]^+$) to these parent $m/z$. More work with a larger number of different compounds and under a larger range of operating conditions (in particular also for different vaporisation/ionisation as well as $MS^2$ fragmentation conditions) is needed in order to test further this assumption and to develop a robust method for the quantification of the relative contributions of these ions to nominal $m/z$ 55 and 57.

### 4.3.2 Differentiation of ions with the same molecular formula, but different molecular structures

While differentiation of ions with the same nominal mass, but different molecular formulas is possible both with the IT-AMS and the ToF-AMS, the differentiation of ions of the same elemental composition, but with different molecular structures is unique to instruments such as the IT-AMS. This feature has already been demonstrated in Sect. 4.2 in the differentiation of singly and doubly charged fragments at $m/z$ 51 in the case of tryptophan, and is further investigated here for ions of the nominal $m/z$ 60 and 73 for a number of carboxylic acids and sugars.

In Fig. 7, results from ToF-AMS measurements and IT-AMS $MS^2$ studies on $m/z$ 60 and 73 are summarised for a variety of sugars (reddish/brownish colours) and carboxylic acids (bluish colours). For $m/z$ 73, also measurements of the polyether PEG200 (pink colour) and of pinonic acid are included in the figure. ToF-AMS measurements indicate the dominant contribution of the ions $[C_2H_4O_2]^+$ and $[C_3H_5O_2]^+$ to $m/z$ 60 and 73, respectively, for both sugars and carboxylic acids (Fig. 7, first column). $MS^2$ measurements with the IT-AMS, however, reveal differences in the molecular structure of these

fragments between the two compound classes, which can be utilised for the differentiation between the latter.

While the ion at $m/z$ 60 from sugars in $MS^2$ experiments fragments only to ions of $m/z$ 42 by loss of water (18 Da), for carboxylic acids additionally a small contribution of the fragment $m/z$ 43 (loss of OH, 17 Da) is observed (Fig. 7a). This can be explained by different molecular structures of the parent ions $[C_2H_4O_2]^+$ for the two compound classes, which can be deduced from the molecular structures of the respective parent compounds (Table 1): For sugars, the fragment ion $[C_2H_4O_2]^+$

most likely consists of a diol, with two OH groups attached to two carbon atoms adjacent to each other. In contrast, in the fragment ion of the same molecular formula formed from fragmentation of carboxylic acids, both oxygen atoms most likely are attached to the same carbon atom, forming a carboxyl group. Alcohols fragment predominantly by loss of water, while

carboxyl groups are known to also fragment by loss of OH (McLafferty and Tureček, 1993), which is in agreement with our findings. For all investigated compounds, ion recovery was found to be 100 %.

Also in MS$^2$ experiments on $m/z$ 73 ([$C_3H_5O_2$]$^+$), differences in the fragmentation chains of carboxylic acids and sugars were observed (Fig. 7b). While $m/z$ 73 from sugars predominantly fragments by loss of CO (28 Da) to ions of $m/z$ 45, for $m/z$ 73 from carboxylic acids this fragmentation pathway is less important, and in contrast it is fragmented predominantly by loss of water (18 Da) to ions of $m/z$ 55. Notably, ion recovery was found to be very different for the two compound groups, with ~100 % ion recovery for carboxylic acids, but only ~60 % for sugars, pointing to an additional fragmentation pathway for $m/z$ 73 from sugars which was not traceable under the given measurement settings. Interestingly, the polyether PEG200 shows a similar fragmentation pattern of $m/z$ 73 as sugars, but with ~100 % ion recovery, pointing to yet another molecular structure of the ion with the chemical formula [$C_3H_5O_2$]$^+$.

These results indicate that with the IT-AMS, it should be possible to determine the relative contributions of carboxylic acids and sugars to the ions $m/z$ 60 and 73 (even though they have the same elemental composition), given no major contributions from other compound classes are present. For MS$^2$ on $m/z$ 60, the ratio of relative intensities of the fragment ions $m/z$ 43 to $m/z$ 42 is a direct indication for the relative contribution of carboxylic acids to the parent ion: the ratio was found to be (0 / 100 %) for the sugars, but in the range (11 ± 1) % / 100 % for the tested carboxylic acids (Fig. 7a). For MS$^2$ on $m/z$ 73, the ratio of relative intensities of the fragment ions $m/z$ 45 to $m/z$ 55 was on average 100 % / 3 % (range 100 % / (2 – 5) %) for sugars, and 32 % / 100 % (range (13 – 57) % / 100 %) for carboxylic acids (Fig. 7b).

Using these results and the average relative contributions of $m/z$ 60 and 73 to the mass spectra of sugars and carboxylic acids, in principle the absolute contribution of these compound classes to the measured total organic mass concentration could be determined from MS$^2$ experiments on ions of $m/z$ 60 and 73, which would not be possible with the regular ToF-AMS. More work is needed in order to test this possibility with a large number of compounds, concentration ranges, range of operating parameters (especially different MS$^2$ fragmentation conditions and vaporisation/ionisation conditions), and also for mixtures of compounds. In first sensitivity studies for MS$^2$ on $m/z$ 73, we already found the presented differentiation to be robust for different aerosol loads, ion quantities in the trapping volume, and ion trapping parameters $q_z$ (see March, 1997). Further characterisation and calibration work is needed in order to fully facilitate quantification of the presented compound classes with the IT-AMS with more robust average values, and to identify the associated limitations and uncertainties, like, e.g., the cross-sensitivity observed for the polyether PEG200 at $m/z$ 73.

## 5 Summary and conclusion

An ion trap aerosol mass spectrometer (IT-AMS) was further developed and optimised, now allowing for more reproducible and reliable measurements while the instrument is also more robust and user-friendly and has extended measurement capabilities implemented in its data acquisition software. This allowed for the instrument's first, 11-day long field deployment, during which it was continuously running in parallel to a ToF-AMS. The results show that the IT-AMS is

capable of providing quantitative information on the major non-refractory submicron species organics, nitrate, and sulphate, with detection limits between 0.5 and 1.4 µg m$^{-3}$ for 1 h-averages. IT-AMS and ToF-AMS apply the same type of ion source (thermal desorption / electron impact ionisation), and consequently the observed organics fragmentation patterns were found to be comparable between the two instruments.

The capability of the IT-AMS to elucidate the structure of the fragmentation products observed in such mass spectra was demonstrated by an MS$^4$ study on tryptophan. Detection limits were found to be sufficiently low to allow for MS$^2$ studies on atmospheric particles under favourable ambient conditions (7 µg m$^{-3}$ at a time resolution of 16 min in the given example), while MS$^{n>2}$ studies are only feasible at higher concentrations. Such a situation can be reached either under laboratory conditions, or via aerosol concentrator systems. All in all, we conclude that the IT-AMS can provide in-depth, quantitative

information on total organic aerosol and on individual ion fragments both in field and in laboratory studies, and that these results are directly relatable to ToF-AMS results.

In laboratory studies on a variety of compounds, we found that the IT-AMS is capable of distinguishing $[C_4H_y]^+$ and $[C_3H_yO]^+$ fragments at the nominal $m/z$ 55 and 57 via their characteristic fragmentation patterns in MS$^2$ experiments. Furthermore, it is possible to distinguish between fragments of the same elemental composition, but with different molecular

structures: different characteristic MS$^2$ fragmentation patterns were found for the ions at $m/z$ 60 and 73 for sugars and carboxylic acids, which have the same elemental compositions ($[C_2H_4O_2]^+$ / $[C_3H_5O_2]^+$) in both cases. While the fragment ions at these $m/z$ originating from carboxylic acids and sugars therefore could not be differentiated with a ToF-AMS, the observed differences in their MS$^2$ fragmentation patterns could be used to distinguish between the two compound classes using the IT-AMS: MS$^2$ at $m/z$ 60 yields ratios of average relative intensities of fragment ions $m/z$ 43 to $m/z$ 42 of

11 % / 100 % for carboxylic acids, but of 0 % / 100 % for sugars. Similarly, at $m/z$ 73, the ratio of relative intensities of the MS$^2$ fragment ions $m/z$ 45 to $m/z$ 55 was on average 100 % / 3 % for sugars, but 32 % / 100 % for carboxylic acids.

More research is needed in order to further constrain these average values by testing a larger number of compounds, and by also testing substances from other compound classes for potential cross-sensitivities. However, these results already indicate that instruments like the IT-AMS potentially enable the distinction between carboxylic acids and sugars in organic aerosol

by means of MS$^2$ experiments, which would further help in organic aerosol characterisation and source apportionment.

**Acknowledgments**

We thank Th. Böttger and the teams of the electronic and the mechanical workshops of the Max Planck Institute for Chemistry for their technical support and valuable help, especially M. Flanz, K.-H. Bückart, and M. Dieterich. We also thank J. Curtius and A. Kürten for their support and technical help regarding the initial design of the IT-AMS. P. Faber is

gratefully acknowledged for providing some of the ToF-AMS data. Funding was provided by the Max Planck Institute for Chemistry and by the Institute for Atmospheric Physics at the Johannes Gutenberg University, Mainz.

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

**Table 1: Organic substances investigated in the laboratory studies. n/a: not available.**

| Compound | Structural formula | Molecular formula | Molecular weight / g mol$^{-1}$ | Purity / % |
|---|---|---|---|---|
| Butyl valerate [a] | | $C_9H_{18}O_2$ | 158.24 | ≥98 |
| Glucose [b] | | $C_6H_{12}O_6$ | 180.16 | >99.5 |
| Glutaric acid [c] | | $C_5H_8O_4$ | 132.12 | 99 |
| Glutathione [b] | | $C_{10}H_{17}N_3O_6S$ | 307.32 | >98 |
| Levoglucosan [d] | | $C_6H_{10}O_5$ | 162.14 | n/a |
| Mannitol [c] | | $C_6H_{14}O_6$ | 182.17 | 99 |
| Oleic acid [e] | | $C_{18}H_{34}O_2$ | 282.47 | n/a |
| PEG200 [c, f] | | $C_{2n}H_{4n+2}O_{n+1}$ | ~200 [g] | n/a |
| *cis*-Pinonic acid [h] | | $C_{10}H_{16}O_3$ | 184.23 | 98 |
| Saccharose [b] | | $C_{12}H_{22}O_{11}$ | 342.30 | >99.5 |
| Succinic acid [c] | | $C_4H_6O_4$ | 118.09 | >99 |
| Tryptophan [b] | | $C_{11}H_{12}N_2O_2$ | 204.23 | n/a |

[a] Sigma-Aldrich Chemie GmbH. [b] Carl Roth GmbH + Co. KG. [c] Alfa Aesar GmbH + Co. KG. [d] Fluka Chemie GmbH. [e] Merck KGaA. [f] Polyethylene glycol. [g] Average molecular mass. [h] Aldrich Chem. Co.

**Table 2: IT-AMS operating parameters in the MS$^n$ studies. $V_{RF}$ is the amplitude (zero-to-peak) of the radiofrequency drive voltage (1.3 MHz) applied to the ring electrode; $q_z$ is the stability parameter resulting from the Mathieu equation (March, 1997). The resonance frequency was experimentally determined.**

| | Trapping phase | | | Reaction phase | | | Resonant excitation | |
| --- | --- | --- | --- | --- | --- | --- | --- | --- |
| $m/z$ | Duration / ms | $V_{RF}$ / V | $q_z$ | Duration / ms | $V_{RF}$ / V | $q_z$ | Amplitude / mV | Frequency / kHz |
| 130 | 50 | 700 | 0.303 | 150 | 700 | 0.303 | 80 | 118.5 |
| 103 | 50 | 700 | 0.383 | 150 | 700 | 0.383 | 80 | 150 |
| 77 | 50 | 700 | 0.512 | 100 | 700 | 0.512 | 120 | 202 |
| 73 | 50 | 258 – 287 | 0.206 – 0.228 | 100 | 402 | 0.320 | 120 | 152.5 |
| 60 | 50 | 258 – 287 | 0.25 – 0.278 | 100 | 402 | 0.389 | 120 | 186.5 |
| 57 | 50 | 258 – 287 | 0.263 – 0.296 | 100 | 402 | 0.409 | 120 | 196.5 |
| 55 | 50 | 258 – 287 | 0.273 – 0.314 | 100 | 402 | 0.424 | 120 | 204.5 |

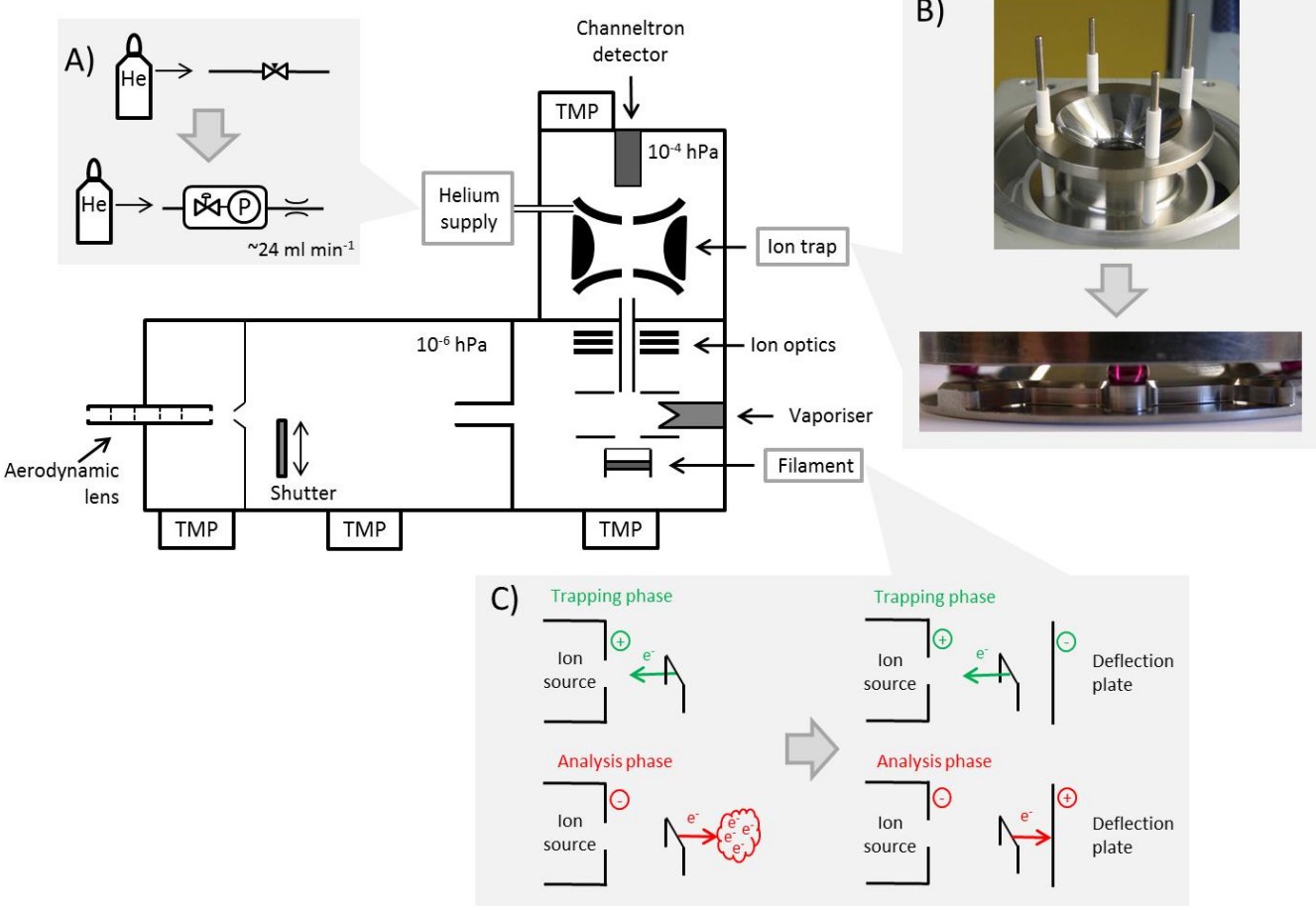

**Figure 1: Schematics of the IT-AMS. The insets A-C visualise the major hardware modifications (compared to Kürten et al., 2007) described in the main text. (TMP: turbo molecular pump; photographs from Fachinger, 2012)**

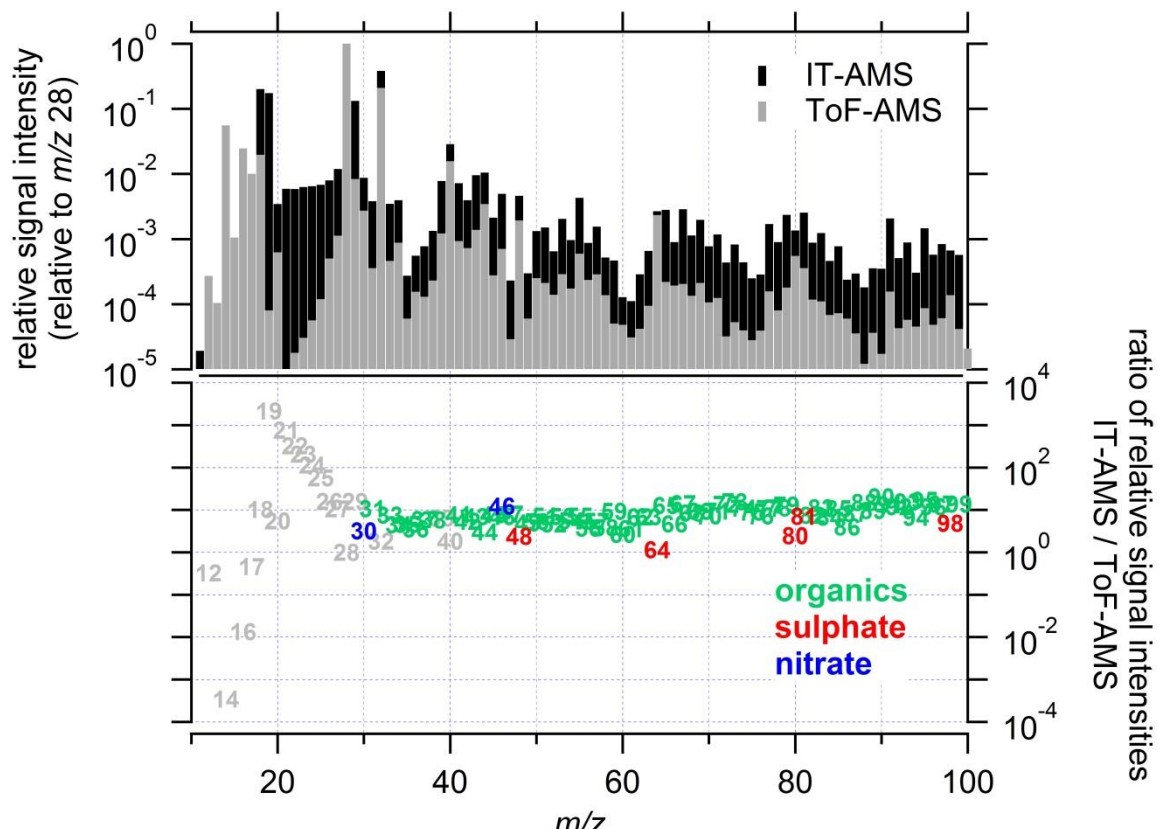

**Figure 2: Comparison of average difference mass spectra measured with the IT-AMS and the ToF-AMS during 11-day long ambient measurements. Shown are the average mass spectra normalised (after conversion to ion rates) to their respective mass spectral signal at *m/z* 28 (upper panel) and the ratio (IT-AMS to ToF-AMS) of these relative signal intensities, colour-coded for the dominant species at the respective *m/z* (lower panel). Note the logarithmic scaling of the y-axes and that the upper panel's y-axis only starts at $10^{-5}$ (i.e., ion signals smaller than that are not shown).**

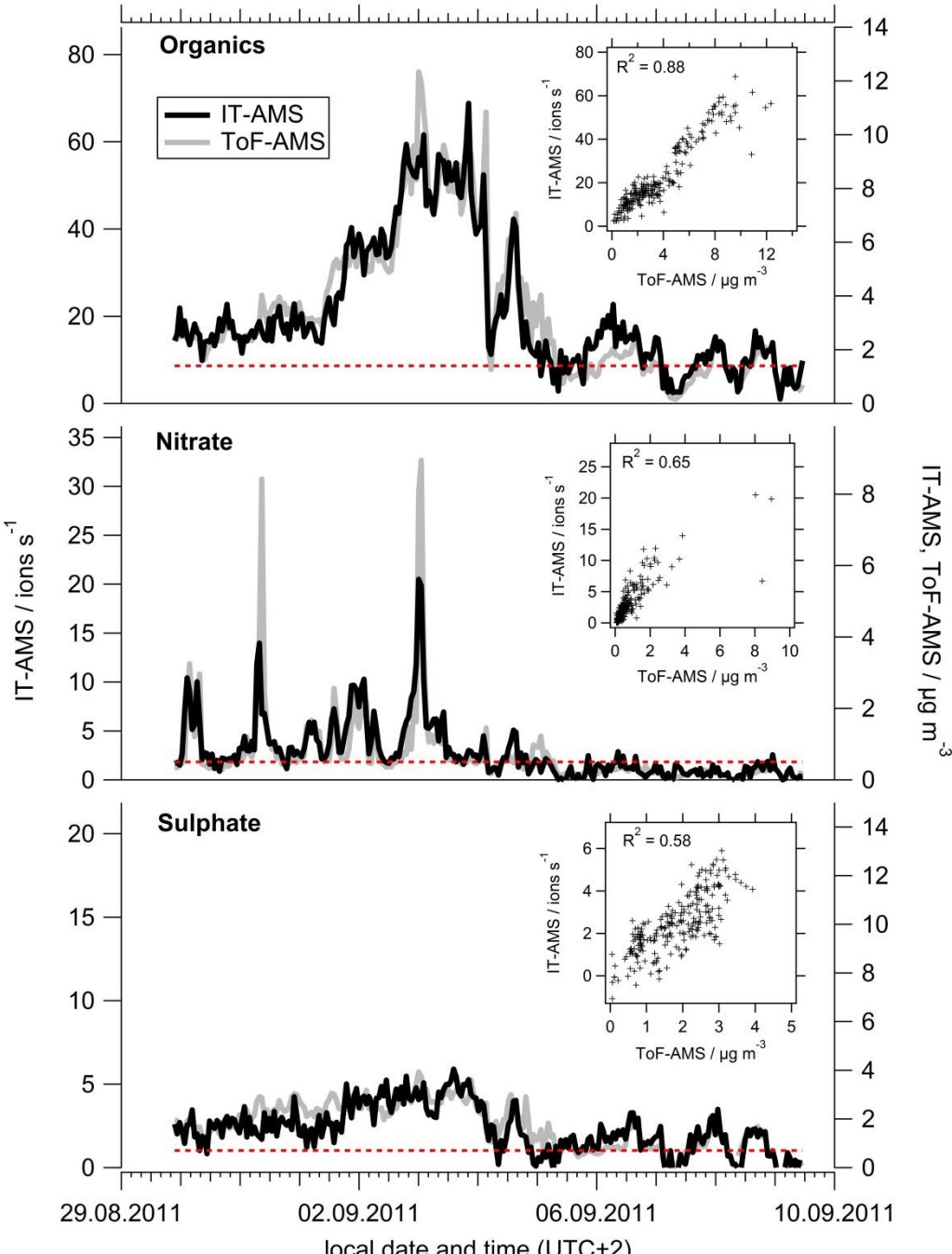

**Figure 3: Comparison of 1 h-averaged time series of organics (upper panel), nitrate (middle panel) and sulphate (lower panel) measured with the IT-AMS (left and right axes for ion rate and mass concentrations, respectively) and the ToF-AMS (right axis) during 11-day long ambient measurements. The inlays show the correlation of the respective time series and the associated squared Pearson's correlation coefficient ($R^2$); the IT-AMS detection limits are marked with red dotted lines. IT-AMS mass concentrations are calculated using the linear correlation with the ToF-AMS time series (inlays).**

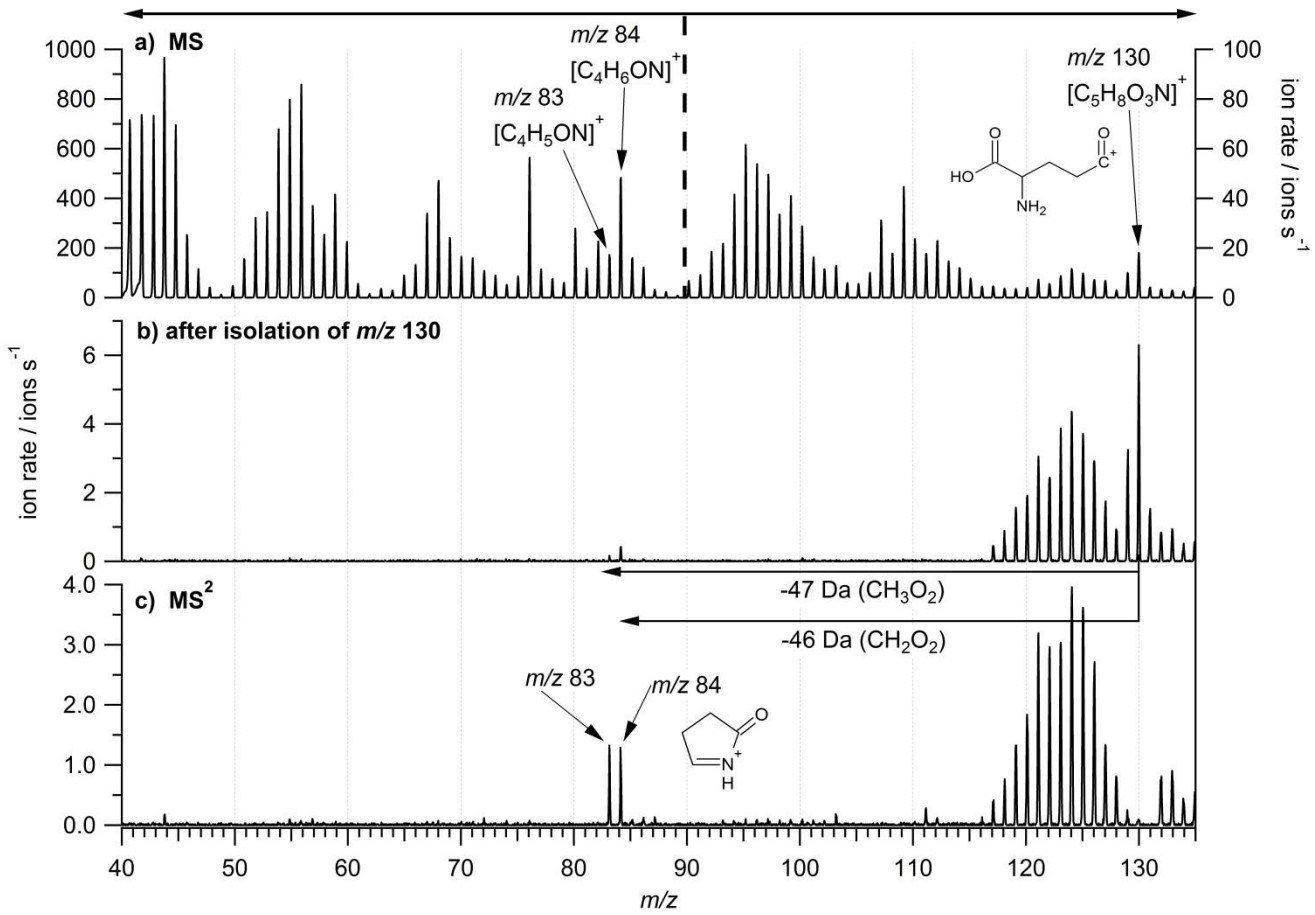

**Figure 4: MS² study on glutathione.** Shown are the "classic" MS (a) and the mass spectrum acquired after isolating (b) and fragmenting (MS², c) *m/z* 130. The signal in c) in the *m/z* range ~115 – 130 originates from ions remaining after the isolation step (b). The dashed vertical line in panel (a) divides the *m/z* ranges for which the left / right y-axes are used, respectively.

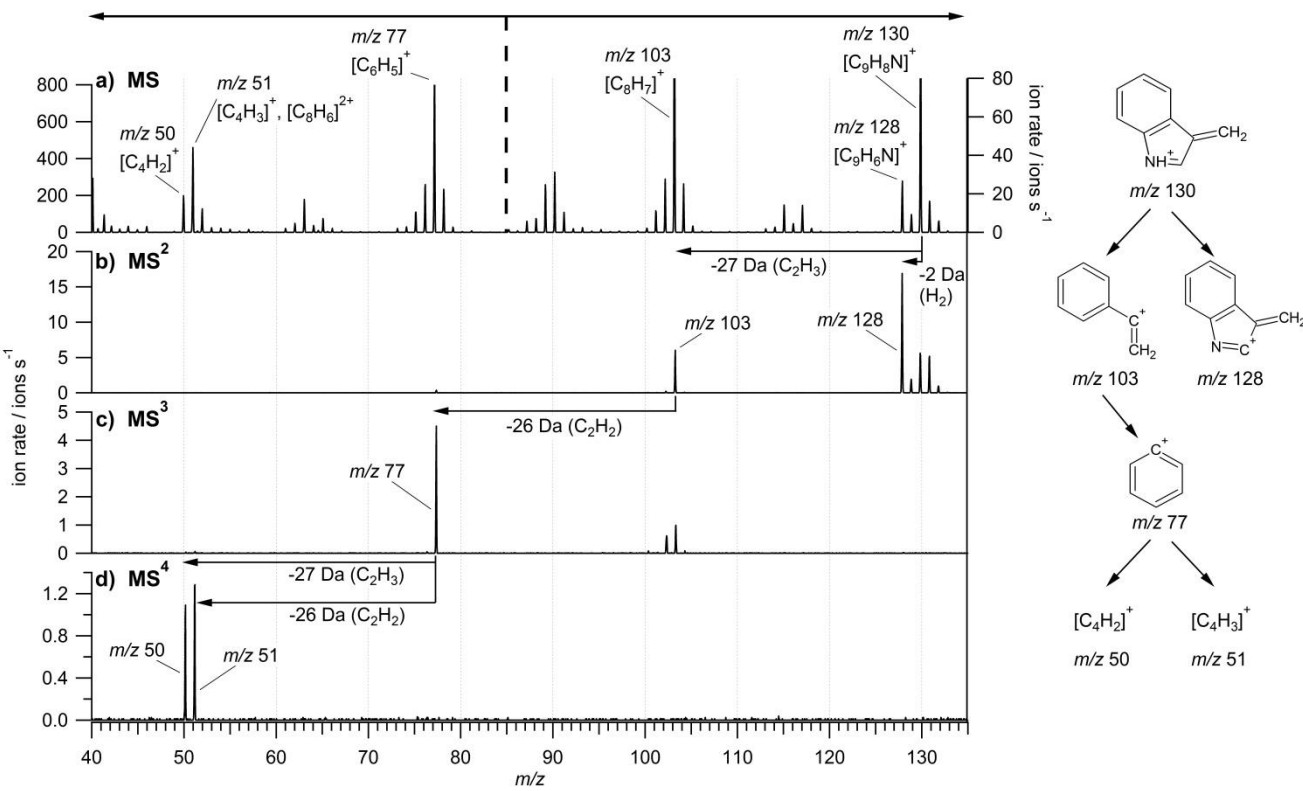

**Figure 5: MS⁴ study on tryptophan. Shown are the "classic" MS (a) and mass spectra acquired after isolating and fragmenting *m/z* 130 (MS², b), *m/z* 103 (MS³, c), and *m/z* 77 (MS⁴, d) from the respective previous step. The dashed vertical line in panel (a) divides the *m/z* ranges for which the left / right y-axes are used, respectively.**

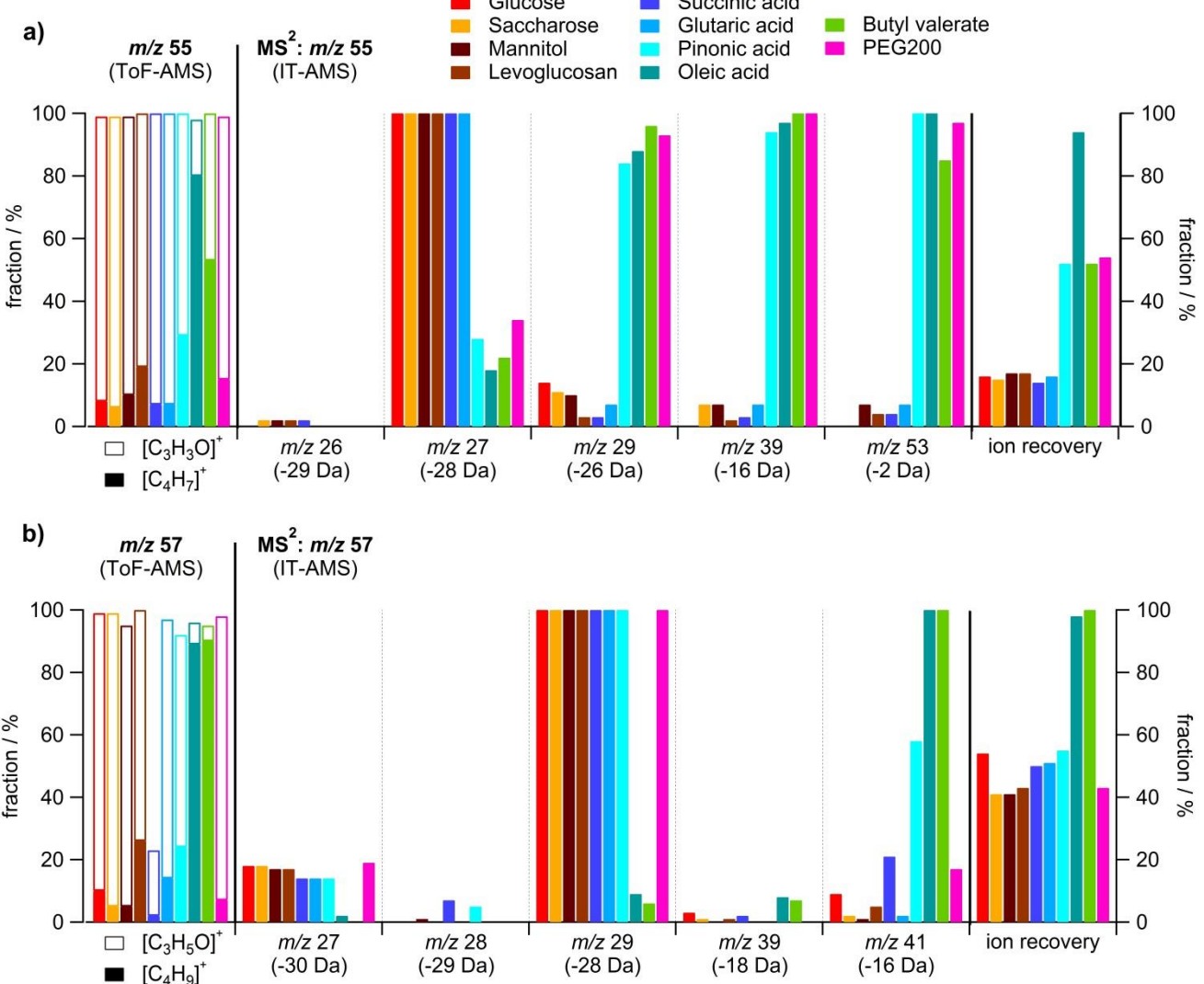

**Figure 6: Results of studies on (a) *m/z* 55 and (b) *m/z* 57 for various compounds. On the left, the relative contributions of [C₃H₃O]⁺**
and [C₄H₇]⁺ to *m/z* 55 and of [C₃H₅O]⁺ and [C₄H₉]⁺ to *m/z* 57 are given (from ToF-AMS measurements; the difference to 100 % is
due to one or several other ions contributing little to the respective *m/z*). On the right, the results from IT-AMS MS² studies on
these ions are summarised. For each investigated compound, the signal intensity of the fragment ions (relative to the signal
intensity of the most abundant fragment) and the ion recovery are given. Not shown are fragments accounting for less than 1 %
and, for the MS² experiment on *m/z* 57, small contributions observed at fragment ion *m/z* 31 for mannitol (1 % relative fraction)
and at *m/z* 42 for glucose, oleic acid, and butyl valerate (3 %, 1 %, and 3 % relative fraction).

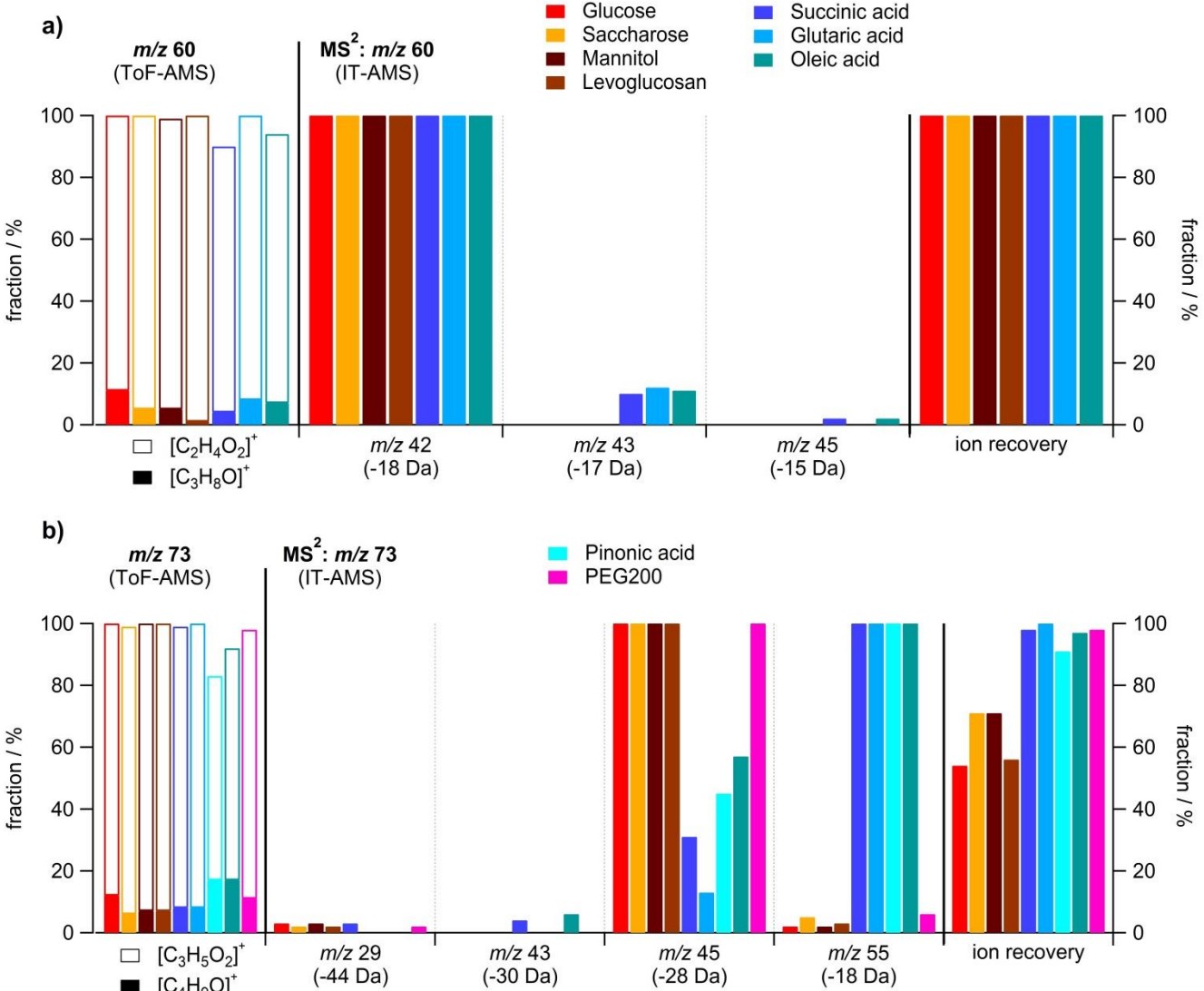

**Figure 7: Results of studies on (a)** *m/z* **60 and (b)** *m/z* **73 for various compounds. On the left, the relative contributions of $[C_2H_4O_2]^+$ and $[C_3H_8O]^+$ to** *m/z* **60 and of $[C_3H_5O_2]^+$ and $[C_4H_9O]^+$ to** *m/z* **73 are given (from ToF-AMS measurements; the difference to 100 % is due to one or several other ions contributing little to the respective** *m/z***). On the right, the results from IT-AMS MS$^2$ studies on these ions are summarised. For each investigated compound, the signal intensity of the fragment ions (relative to the signal intensity of the most abundant fragment) and the ion recovery are given. Not shown are fragments accounting for less than 1 % and, for the MS$^2$ experiment on** *m/z* **73, a small contribution observed at fragment ion** *m/z* **61 for glucose (2 % relative fraction).**