# Peer review of "The ion trap aerosol mass spectrometer: field intercomparison with the ToF-AMS and the capability of differentiating organic compound classes via MS-MS"

_Atmospheric Measurement Techniques, 2016_

## Referee Comment (RC1) · Anonymous Referee #2 · 10 Jan 2017

**General Overview:**

The manuscript, "The ion trap aerosol mass spectrometer: improved design, first field deployment, and the capability of differentiating organic compound classes via MS-MS" by Fachinger et al. provides a useful next step in describing a tandem mass spectrometer (IT-AMS) for analysis of organic aerosols. The portions of the manuscript dealing with $MS^n$ capabilities and the field deployment are of particular interest and offer the reader new information. However, the parts of this manuscript outlining improvements to the instrument first presented by Kürten et al. IJMS 2007, and alluded to in Fachinger (Mainz thesis 2012) do not seem to be significant advances to the instrument or its capabilities to warrant publication in their own right in their current state and should play a more diminished role than suggested by the title of the article. Furthermore, as the Fachinger thesis is published in German it is not accessible to the non-German speaking audience of AMT, and asking the audience to refer to this paper to describe in detail the improvements to the instrument is unhelpful to the reader. In addition, as a thesis has not undergone outside and anonymous peer review, the statements in a thesis cannot be relied upon to the extent used in this manuscript.

The paper claims the IT-AMS is "capable" of quantitative measurements in the field but the instrument was not calibrated in the field using an external standard as to yield a series of mass concentrations over time. The capability of an instrument to be quantitative needs to be backed up with lab or field experiments where the instrument actually is calibrated and determined to be quantitative. It would have been helpful if more information on the field calibration of the IT-AMS and TOF-AMS were given.

There is enough new information in this manuscript however, to support the publication of this manuscript in AMT if the following major corrections are made. I recommend resubmitting the manuscript after concerns with calibration and references to the Fachinger 2012 thesis are addressed.

**Review of content**

**Major concerns:**

**"Improved Design"** – the Kürten et al 2007 paper referenced no need for an improvement in design in order to provide mass concentrations of aerosol components or to become field deployable. To state then in this manuscript that major improvements were needed to make the instrument field deployable leaves the reader questioning the validity of the claim. Of course minor improvements to instrumental design and software modifications normally occur over time but the improvements mentioned could not be published as a stand-alone paper as they do not fundamentally change how the instrument is operated, but only offer minor improvements on existing systems. In my opinion, the "improvements" to the instrument were antidotal unless a direct before after comparison of performance characteristics is given. Furthermore, the Fachinger thesis is asked to be referred to but it is not written in English in which AMT is published. This limits to usefulness of this citation in the following two ways. First a thesis has not undergone scientific peer-review beyond the student's examination committee, and secondly, it is written in German which limits its usefulness to only those AMT readers fluent in both English and

German.  This major concern can be addressed in one of the following ways.  1.  Omit much of the section dealing with the instrument modifications and focus on the TOF-IMS intercomparison and the MS$^n$ data, or 2. expand this section to include a more detailed explanation of the instrument not requiring as much reference to the Fachinger thesis.

**"First field deployment" -** The IT-AMS vs Q-AMS intercomparison for nitrate in Kürten et al 2007 seemed to already be the first field deployment of the IT-AMS instrument although not in its currently modified state.  Furthermore, because the instrument was calibrated it was able to give a mass concentration time series, which was not done in the current manuscript.  Since the IT-AMS had already been field deployed (although perhaps locally in Mainz) and intercomapared to a Q-AMS, the claim here in the title that this is the first field deployment seems overstated.

**Suggestions for the title:**  I suggest rewording the title and refocus the paper to focus less on improvements to the instrument and this being the "first" field deployment and more on the MS$^n$ capabilities and intercomparison with the TOF-AMS.

**Quantification and intercomparison:**  Additional experiments in which the TOF-AMS and the IT-AMS are both calibrated with externally generated aerosol of known mass concentration should be presented.  This type of calibration would validate the IT-AMS calibration method presented in the article as was done for nitrate in the Kürten et al. 2007 paper where both instruments were sampling the same laboratory generated aerosol sample.  The IT-AMS results were thus only given in ion rate which the TOF-AMS results were given in mass concentration ($\mu g/m^3$) but neither seemed to be calibrated in the field.  The reader is left with the question as to why the IT-AMS not was not intercompared with the TOF-AMS in the field using laboratory generated aerosol.  If an intercomparison (similar to that described in the Kürten et al 2007 paper) was conducted, it should be included in the manuscript.  If the intercomparison was not done I would suggest conducting the intercomparison after the fact using the same instrument parameters as when the instruments were in the field.  Not having any field or lab intercomparison so that both instruments are analyzing the same limits the usefulness of the data presented in this manuscript to other readers.  For instance it is important to see if one instrument is reading a higher or lower mass concentration than the other which can offer insight into inlet effects, instrument specific contamination, or bias in one instrument over the other.  Furthermore, presenting IT-AMS only in ion counts makes it difficult for others in the field to compare their data with the data in this manuscript.  Without the intercomparison only the linearity ($R^2$) of the IT-AMS ion rate vs. TOF-AMS mass concentration can be assessed which is done in this manuscript.  But the IT-AMS limit of detection thus relies on the performance of both the IT-AMS and the TOF-AMS which limits the usefulness of this data and suggests to others that a TOF-AMS must be deployed whenever an IT-AMS is deployed in order make quantitative determinations.

**Specific comments pertaining to content:**

**Page 2**

**Section 1: Introduction**

In general -

- One of the benefits of ion traps is the capability of ion/molecule reactions inside the trap which can also be utilized to differentiate isobaric or isomeric ions. This concept should be mentioned as an advantage of ion traps especially since the authors mention the possibility of disadvantageous ion/molecule reactions later in the manuscript.
- The manuscript could be improved by providing references to show the usefulness of differentiating between carboxylic acids and sugars in aerosols.

Line 11: Please be more specific in what types of "mathematical algorithms" have been used.

Line 14: Not sure what "a partial loss of molecular information" is referring to specifically. Do you mean that the ionization could be so complete as to eliminate the molecular ion peak? Or do you mean that hard ionization makes the interpretation of a mixture more difficult? Of course the m/z of the fragments gives much molecular information, however I am guessing the authors probably meant that the molecular ion m/z is lost especially for higher molar mass ions.

Line 13-17 – "On the other hand" - I am not sure what information the author is trying to convey here as EI does not "reduce" complexity of the mass spectrum but increases it compared to soft ionization methods such as chemical ionization. In fact EI of mixtures creates additional complexity which is what the previous sentence addresses requiring the incorporation of mathematical algorithms.

Line 15-16 : "while some important information on the original molecular structures are still retained" - Please state what information is retained. Molecular ion m/z is retained but molecular structure is determined from the fragmentation pattern which in a mixture can be convoluted which requires the mathematical algorithms referred to previously.

Line 26: "differentiation between fragment ions of the same elemental composition, but with different structural formulas." Is there a more concise scientific term for these types of ions such as "isomeric ions" that could be used or defined here?

**Page 3:**

Paragraph 1: I am at a loss as to how a hard ionization technique such as Thermal Desorption (which causes some fragrmentation) coupled to EI which is a "hard" ionization source, could provide "a strong reduction in complexity of organic aerosol mass spectra" compared to soft ionization sources. Does "With these systems" in line 8 refer to the IT-AMS or in fact are you referring to the soft ionization sourced listed before the paragraph shifts to talk of the IT-AMS. Hard ionization produces more fragments and thus more complexity especially for mixtures of molecules. Soft ionization produces less fragments and thus a less complex mass spectra.

Line 10-13:  The Kürten et al. 2007 paper states in its abstract that the instrument at that point was ready for field deployment.   I am not convinced that field deployment was impossible without the modifications to the instrument described in this paper.  This discrepancy needs to be resolved.  Either the instrument was ready for field deployment in the 2007 paper or it wasn't.  As the IT-MS was not calibrated in the field it still may not be considered "field deployable" until it can quantify aerosol components in units of mass concentration.

Line 16 "and potentially quantify"  this term needs to be rephrased as either a technique can quantify or it can't.

**Section 2: Instrumental development**

- It is difficult to tell whether further advances after Fachinger 2012 were made or whether the all of the advances over the Kürten et al. 2007 paper were described by the Fachinger 2012 thesis.
- Also again since Fachinger 2012 was not peer reviewed the reader should not be referred to this thesis multiple times for clarification because the thesis is not peer reviewed and it is not in English, the language of AMT.

**Page 3**

Line 24:  Since the aerodynamic lenses are referenced to Fachinger 2012, were they the same lenses used in the Kürten et al. 2007 paper?  If so, please reference the Kürten paper instead of Fachinger, if not please describe the differences over the lenses in the Kürten paper.

Line 28: Describe to what extent the "flash-vaporization" also causes some ionization or fragmentation before EI causes further ionization and fragmentation.

Line 32:  In reference to "high purity helium" please state the purity and vendor actually used and the vendor.

Line 1:  it is stated the instrument can provide quantitative information as in Kürten et al. 2007 while the last paragraph in the introduction the instrument is only potentially quantitative.

Line 2: Define what you mean here by reproducibility of the measurements.  If the measurements in Kürten et al. 2007 were not reproducible doesn't that call into question the validity of the 2007 paper? Or are you referring to lack of reproducibility as a lack precision in the measurements?

Line 5:  It is stated that the instrument is now more "versatile".  Wasn't the instrument always capable of the measurements made in this paper?  Please explain the instrument now more versatile than it was before?

Line 8-10:  "We only describe the most important changes in detail here; other modifications are only briefly summarized and their details can be found in (Fachinger, 2012)."  Since Fachinger is not peer reviewed and not written in English the details mentioned have not been scientifically reviewed and are

inaccessible to all those who are not bilingual in German and English.  Please either remove most of this section or provide the details here so that they can now be peer-reviewed in English.

Line 11-15:  In the ion trap, instead of the ceramic washers used by Kürten et al. (2007) as spacers between the ring and end capelectrode, now ruby spheres (diameter 6 mm ± 0.635 µm) sitting in precise countersinks (1.3 mm ± 10 µm) are used for a more defined mounting (insert B). This allows for a more reproducible assembly of the electrodes, and consequently more reproducible voltage settings"  To state an improvement the original conditions must also be stated. Please quantify the reproducibility in assembling the electrodes before and after the improvement were made.  Furthermore, ion optics are electrically isolated from each other so that the ring and endcap can voltages can be set independently, how then could imprecision in electrode spacing result in a lack of "reproducible voltage settings". Please provide quantitative proof, or else only state the change without stating that it made an improvement.

Line 15-19:  Please state quantitatively to what extent helium flow changed in the original setup over the course of several days.  Again quantitative evidence is given for the updated system but not for the original system.

Line 18:  State the orifice diameter of the critical orifice.

Line 19: Please state the manufacturer and flow range of the pressure-controlled mass flow controller mentioned.

line 22:  The ion source need not be pulsed if a gate electrode between the ion source and trap is used. Here the ions in the source are continuously produced and the gate electrode voltage is lowered in order to fill the trap, and raised during ion manipulation and scanout to prevent additional ions from entering.  Was this type of operation used and found deficient for some reason?

Line 25:  Was the filament emission current monitored?  If so please quantify the instability.

Line 28: Better reproducibility is mentioned?  State exactly what is more reproducible and how it is measured.

Line 30-34:  Again I suggest that since the specifics are mentioned in Fachinger 2012 that this section describing minor improvements either be expanded on in order to peer review these claims in English. Some of the points like improving the instrument housing, and improving the electrical and communication connections seem like more trivial modifications and can be omitted unless they can be tied back to quantitative instrument improvements.

**Page 5:**

Line 1:  Please state the version of LabVIEW used to write the program.

Line 1-6 – again since Fachinger 2012 is referenced here the nontrivial details referred to need to be provided here.

**Section 3: laboratory and field measurements**

**Page 5**

Line 8:  please include the stated purity and manufacturer of each chemical used.  This could be done in a supplemental table or added to the existing table.

Line 12:  Was the Tof-AMS run in parallel with the IT-AMS so that they were both analyzing the same sample from the nebulizer?  Consider making a schematic diagram of how the instruments were connected together with the nebulizer and/or with the particle counter.

Line 17:  Please mention to what extent each instrument was measuring the same air mass since the measurements seemed to be taken 5 meters apart and temporally (IT-MS 30s time resolution, Tof-AMS 60s time resolution).  Could some of the variability in the instruments mentioned later really be due to the instruments measuring slightly different air masses either spatially or temporally?

Line 19:  How can the instruments be sampling "in parallel" if they were sampling from two separate inlets?  Were the inlets to each instrument both sampling from a common manifold?  Or do you just mean that each instrument was sampling at the same time but from two different (although close) locations.

Also Sobanski et al 2016 doesn't mention the ToF-AMS or the IT-AMS and only refers to instruments in a molile laboratory (MoLa) for measuring aerosol parameters, thus the sentence should be rewritten to so that Sobanski only references the field campaign  and not that "continuous measurements of ambient aerosol using the IT-AMS and a high resolution Tof-AMS were continuously performed" during the campaign.

Line 27 – Kürten et al 2007 states the mass range was up to 200 m/z without using the mass range extension.  Did the mass range decrease over time or did the modifications stated previously lower the mass range?

Line 27-28:  "ions of $m/z$ of interest were isolated (typically within a range of ± 5 $m/z$) by broad band excitation using a filtered noise field"  Figure 4 show isolation was outside of the ± 5 $m/z$ range and looks more like -10 to +5 m/z in that particular instance.  Please revise either your isolation range, or use a different mass spectrum in figure 4.

Line 31:  Please state the versions of IGOR Pro and MATLAB used.

**Page 6**

Line 3:  Please include information on how the "background effects" were performed in the methods section including how the particle free air was generated.

Line 3:  Do you have a quantitative way to state or argue that ionization efficiency in the TOF-AMS didn't change during transport of the instrument to the field location, or over the course of the campaign?  Did you do any field checks of the ionization efficiency?  Please add additional information on how the TOF-AMS was calibrated in Section 2.

Line 4-5:  Again it is vague which "co-located measurements" were taken and how "good agreement" was determined since Fachinger 2012 is not peer reviewed.  Please expand this section.

Line 5-10:  Please explain more fully (or give a reference to)  how detector intensity (in units of voltage or current) is used to calculate an ion rate in units of Hz.

Line 16:  averaging only minimizes the statistical uncertainty if the relative instrumental drift between the two instruments is negligible compared to the variation in the measurements.  What evidence does the authors have in order to quantify instrument drift?  It is stated that 1h average mass spectra "typically" have the same features, but do the atypical results skew the 11day average?

Line 18:  Could the difference in the instrumental response below m/z 30 be due to the gating of the electrons in the IE region of the IT-AMS or different voltage settings between either the IE region and the ion trap or ToF region?  Furthermore could the difference be due to decreased trapping efficiency of low m/z ions.

Line 23: Please quantify "much smaller"

Line 28  Define what it means that "this plateau disappears".  For example, in the plateau disappears then does it then increase like the IT-AMS data?

**Page 6**

Line 3:  Where these calibrations done and accounted for?

Line 4:  The term "is comparible" should be quantified.

Line 6: "is transferable" :  the measurements may be transferrable but since there is still a large variation in the data, transferring the data would cause large uncertainties in the IT-AMS results, and would require that a ToF-AMS always be co-located with a IT-AMS.  This requirement is a major deficiency of the IT-AMS for field campaigns and should be addressed in the manuscript.

Line 9:  is "m/z 48 to 64" supposed to read, "m/z 48 and 64" since only m/z 48 and 64 are designated at sulphates in the upper part of figure 2?

Line 15-16:  "No calibration measurements were performed for the IT-AMS."  This is a major deficiency of the manuscript as this means the IT-AMS mass concentrations must be tied to the ToF-AMS which was also not calibrated in the field.  The authors must explain why neither instrument was not calibrated in the field and intercompared using the same aerosol generation source.

Line 20:  Please comment on the fact that the 1h $R^2$ values for nitrate are lower than that of the 10min data.

Line 24:  It should be noted in the manuscript that Drewnick et al. 2009 were calculating the detection limits for a TOF-AMS and not a IT-MS

Line 25-30:  Using the "linear" relationships in figure 3 to calculate mass concentrations for the IT-AMS will produce large uncertainties in the limit of detection for the IT-AMS (especially for sulphate).  The uncertainty in the detection limits must be calculated and reported.   Also if the $R^2$ value for the 1h measurements of nitrate are larger than the 10min data, how can the limit of detection for the 1hr data be lower than the 10 minute data for nitrate.  Furthermore, if it was claimed in the previous paragraph that the IT-AMS signal for sulfate is lower than that of nitrate, and that sulfate has a lower signal to noise ratio compared to nitrate, how can sulphate and nitrate have the same detection limit?  This discrepancy needs to be explained in the manuscript.

**Section 4.2 and following**

These sections are the highlight of the paper in my opinion and quite well reasoned.

**Section 4.3.2**

The authors make a convincing argument that $MS^2$ studies can differentiate sugars and carboxylic acids for fragment ion isomers.  However, it seems more straight forward to differentiate these species based on their molecular ions in the $MS^1$ spectrum as all of the species in table 1 have different molar masses.  The manuscript could be also be improved by differentiating molecular ion isomers by looking for unique fragments of each isomer, or differences in the ratio of fragments for isomers that do not have unique fragments.

**Section 5**

**Page 12**

line 11-12:  It is stated that that the detection limit "were found to be sufficiently low to allow for $MS^2$ studies on atmospheric particles under favorable ambient conditions (0.6 micrograms/m$^3$)".  The 0.6 figure seems to come not from either ambient measurements or from $MS^2$ measurements.  The 0.6 figure seems to come from the $MS^1$ measurement of tryptophan under laboratory conditions.  I would suggest either referring to the limits of detection for organics, nitrates, and/or sulphates from the field campaign, or the $MS^2$ detection limits from the tryptophan measurements which were stated as being 7 micrograms/m$^3$

**Figure 1:**  Please add the pressure into the ion trap region, and show in which sections turbopumps or other pumps are located.  In subpart "A" show that He comes from a compressed gas cylinder and what the flow rate of He is into the ion trap.  For "B" it looks like these figures are taken directly from Fachinger 2012 and should be referenced as such.

**Figure 2:**  Please make it clear in the text or caption how the relative signal was calculated.  Are these ratios of raw detector output or ratios of ion rates?  Also it is confusing after reading the caption if the top graph is calculated from the relative signals from the bottom graph.  If this is so then the bottom and top graphs should be switched.  I suggest changing the bottom graph's y-axis label to "signals relative to m/z 28.  Also from the caption it is unclear as to how the normalization occurred.  Was the IT-MS data normalized to the m/z 28 value from the IT-AMS, and the ToF-AMS data normalized to the m/z 28 from

the Tof-AMS, or was data from both the IT-AMS, and Tof-AMS normalized to the IT-AMS m/z 28 signal. Also there seems to be some data that is missing such as m/z 13 and 15 that show up in the bottom graph. Below m/z 20 some of the ToF-AMS data is larger than the IT-AMS which obscures the IT-AMS data completely. M/z 21 for the Tof-AMS is not existent in the bottom graph the m/z 21 is shown the top graph. M/z 88 shows a very small signal for the Tof-AMS compared to the IT-AMS signal, but in the top graph m/z 88 has a similar relative signal to m/z 87, and 89. Unless I am reading the graph incorrectly, I am not sure how this discrepancy could occur. Finally, since data below m/z 30 is not useful in differentiating organics, sulphates, or nitrates you may consider omitting this data and only briefly referring to it in the text of the manuscript.

**Figure 3:** For the sulphate graph please align the 0 measurement for the IT-AMS with the 0 measurement of the ToF-AMS like it is for the nitrate and organics graphs. It seems as though some of the data for the IT-AMS might give negative ion rates and thus under the detection limit for the instrument. Please also include a horizontal dotted line on the IT-AMS graph indicating the calculated limit of detection of the instrument. It looks like the data for nitrates after September 8[th] might also be below the detection limit, so this should be discussed in the text. Finally, local date and time, should be designated in the caption vs UTC so that the reader knows the time zone and if local time includes daylight savings time.

**Figure 4.** The vertical dotted line in graph a) should be defined in the caption. It looks like the dotted line is used to differentiate a region to the left which uses the left y-axis, and a region to the right that uses a different y-axis. Please include only a graph of the isolation step before graph b).

**Specific comments pertaining to format:**

**Page 2**

Line 7: **"**The currently most widely…" suggested rephrase to "Currently, the most widely"

line 10: delete "here" and change "is" to "are" in the sentence "Since here a large number of different molecules is analysed simultaneously"

Line 13: Suggest changing "strongly fragmented" to "highly fragmented"

Line 18: "high resolution" to "higher resolution" as high resolution mass spectrometry often refers to Penning trap instruments such as ICR or Orbitrap which have resolutions orders of magnitude greater than TOF. It would also be good to state here what the resolution of IT-AMS is from the Kürten et al. 2007 paper.

Line 25: consider removing "some" in "While some information on elemental composition".

**Page 6**

line 15:  Please rephrase this sentence so that it doesn't end in "with."

**Page 12**

line 5: remove the word "regular" in "regular ToF-AMS

page 12 line 30:   replace "on" with "in"

**References**

Fachinger, J. R. W.: Das Aerosol-Ionenfallen-Massenspektrometer (AIMS): Aufbau, Charakterisierung und Feldeinsatz, PhD thesis, Johannes Gutenberg University Mainz, Mainz, 2012.

Kürten, A., Curtius, J., Helleis, F., Lovejoy, E. R., and Borrmann, S.: Development and characterization of an ion trap mass spectrometer for the on-line chemical analysis of atmospheric aerosol particles, International Journal of Mass Spectrometry, 265, 30-39, 2007.

Sobanski, N., Tang, M. J., Thieser, J., Schuster, G., Pöhler, D., Fischer, H., Song, W., Sauvage, C., 5 Williams, J., Fachinger, J., Berkes, F., Hoor, P., Platt, U., Lelieveld, J., and Crowley, J. N.: Chemical and meteorological influences on the lifetime of NO3 at a semi-rural mountain site during PARADE, Atmos. Chem. Phys., 16, 4867-4883, 2016.

---

## Referee Comment (RC2) · Anonymous Referee #1 · 26 Jan 2017

The manuscript by Fachinger et al. describes technical improvements applied to a previously developed ion trap aerosol mass spectrometer (IT-AMS). The IT-AMS measures the chemical composition of aerosol particles by means of flash vaporization followed by electron impact ionization and ion analysis with an IT mass spectrometer. While the mass spectrometer and ion detection unit were developed in-house, the vacuum chamber with its aerodynamic lens inlet and the vaporization/ionization unit are identical to the Aerodyne AMS vacuum chamber. The most commonly used commercially available aerosol mass spectrometers nowadays use time of flight (ToF) mass spectrometers, which e.g. allow to derive information on the origin of the organic fraction of the aerosol constituents due to the high mass resolving power of a ToF. However, an IT has the capability of performing so-called $MS^n$ studies, which can help to distinguish between fragments with different isomeric structures. Therefore, the IT-AMS has, in principle, an important advantage over other commercially available aerosol mass spectrometers.

The capabilities of the IT-AMS are demonstrated by lab experiments, where generated organic particles are analyzed by $MS^n$ studies. The different fragmentation patterns show clear differences between different compound classes. Furthermore, measurements during a field campaign indicate very similar results for the IT-AMS and a ToF-AMS for the nitrate, organic and sulfate fraction of ambient aerosols.

Regarding the description of the modifications, I agree with referee #2 that these need to be explained in much greater detail. Just referring to a German PhD thesis is not sufficient, especially since the chosen journal (AMT) is actually well-suited for a description of technical details that improve a measurement technique.

However, overall the paper is well-written and should be published in AMT after addressing the comments listed in the following as well as the requested improvement on describing technical details.
* * *
Specific comments:

Page 1, line 19: change „was demonstrated" to "is demonstrated"

Page 2, line 3: the paper by Schramm et al. (2009) should also be cited in this respect, especially since the paper makes uses of a similar ion trap as used in the present study

Page 4, line 12: please mention how many ruby spheres were used

Page 4, line 27: please explain what "modified" exactly means

Page 4, line 29: the reason for using the deflection plate and its functionality should be discussed

Page 5, line 2/3: please explain how the semi-automatic tuning of operation parameters works

Page 5, line 26: up to what m/z can the mass range be extended? Up to what m/z was the set-up tested?

Page 6, line 9/10: what about other compounds like water, ammonium and chloride? If the usable mass range starts at 30 amu, it means that the important compound class ammonium cannot be measured as in the standard AMS, please discuss

Page 7, line 1: should there be a "≥" sign instead of ">"?

Page 7, line 20-23: this cannot be the only explanation as the signals are well above the LOD (1.3 / 0.7 μg/m3), any other ideas?

Page 8, line 26: what is the fraction of the doubly charged ions? where are they coming from (from the ion source or from reactions inside the ion trap)?

Page 9, line 24: please explain better how the value of the ion recovery is exactly determined

Page 10, line 27: please add "and pinonic acid" after „… (pink color)"

Figure 1: (i) the photographs in panel b) are too small, (ii) the functionality of the deflection plate shown in panel c) should be explained in more detail in the text

Figure 2: is this the final data after correcting for the backgrounds, etc.? it says the IT-AMS has a lower signal to noise, but why are the IT-AMS signals larger by a factor of 10? if this is the case then the sensitivity could be significantly improved by reducing the noise; what is causing the high noise?

Figure 3: sulfate: memory effects? heater temp. the same?

Figure 6 and Figure 7: explain what "fraction" means as the fragments do not seem to add up to 100%

--

References:

Schramm, E., et al.: Trace Detection of Organic Compounds in Complex Sample Matrixes by Single Photon Ionization Ion Trap Mass Spectrometry: Real-Time Detection of Security-Relevant Compounds and Online Analysis of the Coffee-Roasting Process, *Anal. Chem.*, 81(11), 4456–4467, DOI: 10.1021/ac900289r, 2009.

---

## Author Comment (AC1) · 8 Mar 2017

The comment was uploaded in the form of a supplement:
http://www.atmos-meas-tech-discuss.net/amt-2016-370/amt-2016-370-AC1-supplement.zip

---

## Author Response (AR1)

**Reply to reviews**

Note: all Page and Line numbers in our reply refer to the revised manuscript showing track changes.

**Referee #1:**

General Overview:

The manuscript, "The ion trap aerosol mass spectrometer: improved design, first field deployment, and the capability of differentiating organic compound classes via MS-MS" by Fachinger et al. provides a useful next step in describing a tandem mass spectrometer (IT-AMS) for analysis of organic aerosols. The portions of the manuscript dealing with MSn capabilities and the field deployment are of particular interest and offer the reader new information. However, the parts of this manuscript outlining improvements to the instrument first presented by Kürten et al. IJMS 2007, and alluded to in Fachinger (Mainz thesis 2012) do not seem to be significant advances to the instrument or its capabilities to warrant publication in their own right in their current state and should play a more diminished role than suggested by the title of the article. Furthermore, as the Fachinger thesis is published in German it is not accessible to the non-German speaking audience of AMT, and asking the audience to refer to this paper to describe in detail the improvements to the instrument is unhelpful to the reader. In addition, as a thesis has not undergone outside and anonymous peer review, the statements in a thesis cannot be relied upon to the extent used in this manuscript.

The paper claims the IT-AMS is "capable" of quantitative measurements in the field but the instrument was not calibrated in the field using an external standard as to yield a series of mass concentrations over time. The capability of an instrument to be quantitative needs to be backed up with lab or field experiments where the instrument actually is calibrated and determined to be quantitative. It would have been helpful if more information on the field calibration of the IT-AMS and TOF-AMS were given.

There is enough new information in this manuscript however, to support the publication of this manuscript in AMT if the following major corrections are made. I recommend resubmitting the manuscript after concerns with calibration and references to the Fachinger 2012 thesis are addressed.

We thank the reviewer for his/her very thoughtful review and the valuable and constructive comments. The reasoning behind omitting some technical details and referring to the PhD thesis instead was to ensure easier readability. However, as the reviewer correctly points out, the paper should be able to be understood by itself, without needing to refer to the PhD thesis. Therefore, we added more technical details to the revised version instead of referring to the thesis, as detailed in the replies to the specific comments below.

Regarding the calibration, we do not completely agree with the reviewer's comment. Kürten et al. (2007) already demonstrated the capability of the IT-AMS to quantify nitrate mass concentrations after performing laboratory calibrations using pure ammonium nitrate particles. In our work the IT-

AMS instead was calibrated in the field by comparison to the ToF-AMS data. We do not see the fundamental disadvantage in calibrating an instrument by comparison with an independently calibrated similar instrument (here: the ToF-AMS) compared to a calibration by indirect comparison with data calculated from a completely different instrument (e.g. mass concentrations calculated from CPC data and particle sizes determined with a DMA). The applied method also adds valuable new information on relative ionization efficiencies of organics and sulphate, as discussed in the replies to the specific comments below; in light of the reviewer's comment, we added this discussion also in the revised manuscript. We also added more details on the calibration of the reference instrument (the ToF-AMS) and on how the IT-AMS was calibrated using the ToF-AMS. See also the replies to the specific comments below for further details.

We thoroughly revised the whole manuscript according to the detailed reviewer's comments (see below), and hope to thereby have cleared all of the reviewer's concerns.

Review of content

Major concerns:

"Improved Design" – the Kürten et al 2007 paper referenced no need for an improvement in design in order to provide mass concentrations of aerosol components or to become field deployable. To state then in this manuscript that major improvements were needed to make the instrument field deployable leaves the reader questioning the validity of the claim. Of course minor improvements to instrumental design and software modifications normally occur over time but the improvements mentioned could not be published as a stand-alone paper as they do not fundamentally change how the instrument is operated, but only offer minor improvements on existing systems. In my opinion, the "improvements" to the instrument were antidotal unless a direct before after comparison of performance characteristics is given. Furthermore, the Fachinger thesis is asked to be referred to but it is not written in English in which AMT is published. This limits to usefulness of this citation in the following two ways. First a thesis has not undergone scientific peer-review beyond the student's examination committee, and secondly, it is written in German which limits its usefulness to only those AMT readers fluent in both English and German. This major concern can be addressed in one of the following ways. 1. Omit much of the section dealing with the instrument modifications and focus on the TOF-IMS intercomparison and the MSn data, or 2. expand this section to include a more detailed explanation of the instrument not requiring as much reference to the Fachinger thesis.

While the instrument as described by Kürten et al. (2007) was running stable in the laboratory and in principle would have been field deployable (in the sense of providing aerosol mass concentrations also in the field), it would have been a major effort to put it in the field and get it to run in a limited amount of time, as typical for a measurement campaign. The modifications made to the instrument were indispensable in order for the instrument to be set up in a reasonable amount of time, and from the get-go running stable without the need for further tuning etc. Therefore we think the statement that only the modifications allowed for a field deployment is justified, as setting an instrument up in a reasonable amount of time is one of the major criteria in order to determine whether an instrument is really "fit" to be field-deployed. However, also in light of some other comments by the reviewer, we changed the title to put the focus less on the modifications and the field deployment, and more to the overall comparability to the ToF-AMS (see reply to comment

below). Although they might not justify a stand-alone paper, we think it is important and well within the scope of AMT to report the technical modifications made to the instrument described by Kürten et al. (2007), as also pointed out by referee #2. In order to avoid referencing the Fachinger (2012) thesis, we added more technical details on these modifications in several locations, see replies to comments below.

"First field deployment" - The IT-AMS vs Q-AMS intercomparison for nitrate in Kürten et al 2007 seemed to already be the first field deployment of the IT-AMS instrument although not in its currently modified state. Furthermore, because the instrument was calibrated it was able to give a mass concentration time series, which was not done in the current manuscript. Since the IT-AMS had already been field deployed (although perhaps locally in Mainz) and intercomapred to a Q-AMS, the claim here in the title that this is the first field deployment seems overstated.

While it is true that the IT-AMS in its previous version (as described by Kürten et al., 2007) already measured ambient aerosol by fitting an inlet line from outside the laboratory to the instrument, it was not deployed outside the laboratory (i.e., deployed after moving it from the lab to a different place – the "field", getting it running again in a limited amount of time, under non-laboratory conditions, etc. – i.e., all the procedures associated with a typical field measurement campaign outside the "home" laboratory). So indeed the instrument in its current state for the first time was deployed in the field (i.e., outside the laboratory), which was only possible with reasonable effort due to the technical improvements described in the manuscript. However, also in light of the next comment, we changed the title to focus more on the comparability with the ToF-AMS instead of the instrumental modifications (see reply to next comment).

Suggestions for the title: I suggest rewording the title and refocus the paper to focus less on improvements to the instrument and this being the "first" field deployment and more on the MSn capabilities and intercomparison with the TOF-AMS.

We changed the title to "The ion trap aerosol mass spectrometer: field intercomparison with the ToF-AMS and the capability of differentiating organic compound classes via MS-MS" in order to have it focusing less on the technical modifications, as suggested by the reviewer.

Quantification and intercomparison: Additional experiments in which the TOF-AMS and the IT-AMS are both calibrated with externally generated aerosol of known mass concentration should be presented. This type of calibration would validate the IT-AMS calibration method presented in the article as was done for nitrate in the Kürten et al. 2007 paper where both instruments were sampling the same laboratory generated aerosol sample. The IT-AMS results were thus only given in ion rate which the TOF-AMS results were given in mass concentration (µg/m3) but neither seemed to be calibrated in the field. The reader is left with the question as to why the IT-AMS not was not intercompared with the TOF-AMS in the field using laboratory generated aerosol. If an intercomparison (similar to that described in the Kürten et al 2007 paper) was conducted, it should be included in the manuscript. If the intercomparison was not done I would suggest conducting the intercomparison after the fact using the same instrument parameters as when the instruments were in the field. Not having any field or lab intercomparison so that both instruments are analyzing the same limits the usefulness of the data presented in this manuscript to other readers. For instance it is important to see if one instrument is reading a higher or lower mass concentration than the other which can offer insight into inlet effects, instrument specific contamination, or bias in one instrument over the other. Furthermore, presenting IT-AMS only in ion counts makes it difficult for others in the

field to compare their data with the data in this manuscript. Without the intercomparison only the linearity (R2) of the IT-AMS ion rate vs. TOF-AMS mass concentration can be assessed which is done in this manuscript. But the IT-AMS limit of detection thus relies on the performance of both the IT-AMS and the TOF-AMS which limits the usefulness of this data and suggests to others that a TOF-AMS must be deployed whenever an IT-AMS is deployed in order make quantitative determinations.

Calibration of the IT-AMS against mass concentrations determined by the independently carefully calibrated ToF-AMS is not fundamentally different to a calibration of the IT-AMS against mass concentrations determined from other measurements like CPC concentrations of size-resolved laboratory particles. This is identical to the regular practice for calibration of several kinds of aerosol instruments and other measurement instruments by manufacturers. The main difference is that this kind of indirect calibration has the potential for slightly increased uncertainty compared to a direct calibration using a standard. We therefore do not agree with the reviewer that the calibration against the ToF-AMS, as performed in this work, limits the usefulness of the insights gained from the comparison, especially since we have shown that there was no temporal variability in the relationship between the data of both instruments.

Furthermore, it was already shown by Kürten et al. (2007) for both sulfate and nitrate that the IT-AMS responds linearly to changes in mass concentration. It was also shown by Kürten et al. (2007) that independent calibration of IT-AMS and Q-AMS with ammonium nitrate leads to comparable nitrate mass concentrations obtained from both instruments in ambient measurements. Therefore, the overall comparability of the IT-AMS to Q-AMS (and therefore also ToF-AMS) mass concentration readings is already validated.

In this work, we calibrate the IT-AMS not only for nitrate, but also for sulfate and organics using the independently calibrated ToF-AMS mass concentrations. By this means, we can calculate relative ionization efficiency (RIE) values for the IT-AMS for both sulfate and organics with respect to nitrate. We obtain values of 0.4 for sulfate, and 1.7 for organics. This slightly higher value for organics compared to the one used for the ToF-AMS is in line with the slightly better ion transmission efficiency at higher m/z for the IT-AMS, as described in Sect. 4.1 of the manuscript. For sulfate, an RIE value of 0.4 is found, which is significantly smaller than the RIE value of the ToF-AMS (1.2). This is in line with the depletion of the sulfate fragment ions within the IT-AMS, as also discussed in Sect. 4.1. While of course these RIE values can be further constrained and improved in additional measurements, they already give a good first estimate of RIE values needed when the IT-AMS is calibrated with ammonium nitrate, like it is regularly done for the ToF-AMS. We therefore think that the comparison with the ToF-AMS gives valuable new insight into the response of the IT-AMS to the major species apart from nitrate (i.e., sulfate and organics). We have included this discussion on RIE values in the revised version of the manuscript (P8 L32-P10 L12), also in line with the requested stronger focus on the IT-AMS / ToF-AMS intercomparison:

"From the calibration of the IT-AMS against the ToF-AMS, furthermore relative ionisation efficiency (RIE) values for sulphate and organics can be calculated for the IT-AMS. The relative ionisation efficiency is a constant factor with which the ionisation efficiency (determined in calibrations using ammonium nitrate) is multiplied in order to get the species-dependent ionisation efficiency. In order to determine these RIEs, the slope obtained for nitrate from the correlation depicted in Fig. 3 (inlay) is related to those determined for organics and sulphate, respectively. By this means, RIE values of 0.4 for sulphate and of 1.7 for organics were found. For organics, this is slightly higher than the RIE

value of 1.4 used for the ToF-AMS, consistent with the slightly higher ion transmission of the IT-AMS for larger *m/z* (to which mostly organics are contributing). For sulphate, the RIE of 0.4 is much smaller than the sulphate RIE value typically used for the ToF-AMS (1.2), consistent with the depletion of sulphate-related ions in the IT-AMS, as described above. It also has to be kept in mind that even without those influences, not exactly the same RIE values as for the ToF-AMS can be expected due to the use of a simplified fragmentation pattern table (see Sect. 3).

By calibrating the IT-AMS for nitrate as demonstrated by Kürten et al. (2007), with the RIE values determined here the mass concentrations of sulphate and organics can be directly obtained from IT-AMS measurements. Note, however, that RIE values might change with different accumulation and reaction times, and therefore need to be newly measured when changing the instrumental settings."

In order for the data to be comparable to other measurements, we also converted the ion rates given in Fig. 3 to mass concentrations using the calibration against the ToF-AMS. We now report both ion rates and absolute mass concentrations of the IT-AMS for the three species organics, nitrate and sulfate in Fig. 3. The mass concentrations derived for the IT-AMS (corrected for ammonium and black carbon) within the uncertainty agree well within those of independent PM1 measurements, further validating the calibration against the ToF-AMS, as we now also discuss (P9, L19-24):

"The mass concentration time series derived from the IT-AMS measurements using the linear correlation with the ToF-AMS measurement (Fig. 3, inlays) within their uncertainty agree well with those of co-located measurements: the sum of black carbon (from MAAP) with IT-AMS sulphate, nitrate, and organics and corrected for the missing species ammonium by assuming fully neutralised aerosol (as expected for regional background aerosol and validated by the ToF-AMS measurements) correlates well with the total $PM_1$ mass concentration measured with the EDM (slope = 1.03, $R^2$ = 0.64 for 1 h data)."

We also want to stress that the reviewer's statement "neither seemed to be calibrated in the field" is not correct. The ToF-AMS was calibrated directly prior to the campaign, and as discussed in response to some comments below, we are very confident that this calibration is valid throughout the field measurement, so the ToF-AMS indeed serves as a valid, calibrated reference instrument.

We therefore also do not agree with the statement that "the IT-AMS limit of detection thus relies on the performance of both the IT-AMS and the TOF-AMS which limits the usefulness of this data and suggests to others that a TOF-AMS must be deployed whenever an IT-AMS is deployed in order make quantitative determinations." The ToF-AMS here serves as reference instrument, and the IT-AMS data are calibrated using the ToF-AMS data. As we have shown in the manuscript, this calibration using the ToF-AMS is stable over time, making the once calibrated IT-AMS an instrument which can operate as independently as any other AMS. Of course the ToF-AMS data come with some uncertainty, but so would data acquired in other independent measurements like mass concentration calculated from a CPC measurement of size-selected particles of known composition. However, we agree that the uncertainty of the derived IT-AMS mass concentrations needs to be stated, and have included this in the manuscript (see reply to comment below). With the results from this work, it is possible to quantify organic and sulphate mass concentrations in addition to nitrate mass concentrations, by only calibrating with ammonium nitrate and using RIE values for sulphate and organics, as done regularly for the ToF-AMS. There is no question that additional measurements

with laboratory-generated aerosol of various substances (as suggested by the reviewer) are desirable in the future to further constrain the RIE values determined here. This however should be done as a function of IT-AMS instrument settings and would be a whole study in itself and is outside the scope of the present manuscript.

- One of the benefits of ion traps is the capability of ion/molecule reactions inside the trap which can also be utilized to differentiate isobaric or isomeric ions. This concept should be mentioned as an advantage of ion traps especially since the authors mention the possibility of disadvantageous ion/molecule reactions later in the manuscript.
- The manuscript could be improved by providing references to show the usefulness of differentiating between carboxylic acids and sugars in aerosols.

We added the mention of ion/molecule reactions in ion traps for structure determination to the introduction (P3, L1-2):

"Additionally, ion/molecule reactions inside the ion trap can be utilized in order to differentiate between isobaric or isomeric ions (e.g., Kascheres and Cooks, 1988)."

We also added a general statement on the usefulness of the differentiation of sugars and carboxylic acids for source apportionment, including a reference regarding the different sources of sugars and carboxylic acids in atmospheric aerosol (P3, L21-24):

"Since sugars and carboxylic acids can be associated with different aerosol sources (sugars originate e.g. from biomass burning or primary biological material, while carboxylic acids originate e.g. from photo-oxidation of organic precursors (Graham et al., 2003)), this would help in further improving the differentiation of various organic aerosol components and therefore in source apportionment of atmospheric organic aerosol."

Line 11: Please be more specific in what types of "mathematical algorithms" have been used.

This is specified in the provided reference (Allan et al., 2004): different species contributing to ambient aerosol are deconvoluted according to their typical fragmentation pattern. We changed "mathematical algorithms" to "a mathematical deconvolution algorithm" to be more specific; additional information can be found in the provided reference.

Line 14: Not sure what "a partial loss of molecular information" is referring to specifically. Do you mean that the ionization could be so complete as to eliminate the molecular ion peak? Or do you mean that hard ionization makes the interpretation of a mixture more difficult? Of course the m/z of the fragments gives much molecular information, however I am guessing the authors probably meant that the molecular ion m/z is lost especially for higher molar mass ions.

Yes, we refer to the fact that fragmentation is very strong and therefore the molecular ion as well as other larger fragment ions typically are not present in the mass spectrum. We clarified by adding:

"(e.g., typically no molecular ion is observed)" (P2, L14-15)

Line 13-17 – "On the other hand" - I am not sure what information the author is trying to convey here as EI does not "reduce" complexity of the mass spectrum but increases it compared to soft ionization methods such as chemical ionization. In fact EI of mixtures creates additional complexity which is what the previous sentence addresses requiring the incorporation of mathematical algorithms.

Yes and no – this depends on the perspective from which one is looking at it. We refer here to mixtures of many different organic compounds as typical for ambient organic aerosols, not to the mass spectrum of one individual compound. To simplify, the EI mass spectrum of a single compound is more complex (i.e., contains a much larger number of fragments) than a mass spectrum obtained after soft ionization, which is what the reviewer is referring to. However, when looking at atmospheric mixtures of organic molecules, soft ionization will (in the extreme case) produce a molecular ion of each single type of molecule, which (again, oversimplified and exaggerated) will result in a single m/z line for each of the thousands of molecules, which contains a lot of information but can be hard to interpret. In contrast, EI reduces very different complex molecules to typical, common fragment ions, so while the mass spectrum of a single compound using EI might be more complex, the mass spectrum of a mixture of thousands of different molecules is much simpler to interpret (however, can be interpreted only to a lower degree due to the loss of molecular information) than such a mass spectrum from soft ionization, since the information is already reduced to the "common" fragments (i.e., some molecular features the different types of molecules have in common). We clarified this by adding (P2, L15-16):

"On the other hand, since different molecules containing the same sub-structure will be reduced to the same fragment ions, …"

Line 15-16 : "while some important information on the original molecular structures are still retained" - Please state what information is retained. Molecular ion m/z is retained but molecular structure is determined from the fragmentation pattern which in a mixture can be convoluted which requires the mathematical algorithms referred to previously.

We hope this is now clarified by our replies to the two previous comments. We are not referring to the molecular ion here, but to the fragment ions which still contain some information on the structure of the original molecule. We clarified this in the revision (P2, L18):

"…while some important information on the original molecular structures are still retained in these common fragment ions,…"

Line 26: "differentiation between fragment ions of the same elemental composition, but with different structural formulas." Is there a more concise scientific term for these types of ions such as "isomeric ions" that could be used or defined here?

We added the definition "(i.e., isomeric ions)" to the sentence.

Page 3:

Paragraph 1: I am at a loss as to how a hard ionization technique such as Thermal Desorption (which causes some fragrmentation) coupled to EI which is a "hard" ionization source, could provide "a strong reduction in complexity of organic aerosol mass spectra" compared to soft ionization sources. Does "With these systems" in line 8 refer to the IT-AMS or in fact are you referring to the soft ionization sourced listed before the paragraph shifts to talk of the IT-AMS. Hard ionization produces more fragments and thus more complexity especially for mixtures of molecules. Soft ionization produces less fragments and thus a less complex mass spectra.

As already discussed in the reply to a similar comment above, this pretty much depends on the point of view under which this is looked at and on the number of different species within the sample. Indeed, as the reviewer states, much more fragmentation occurs in the "harder" desorption/ionization technique as used in the IT-AMS than with "soft" ionization techniques. Therefore, the EI mass spectrum obtained from a single compound will be much more "complex" than one obtained from a soft ionization technique, where ideally only the molecular ions are visible. However, when dealing with atmospheric aerosol, such might not be desirable (at least not in a one-dimensional analysis technique): the thousands of different molecules present in ambient aerosol potentially all would give a different molecular ion signal. This means that in order to characterize ambient organic aerosol calibration measurements for a large number of different chemical compounds might be needed. If using a "harder" ionization technique, these molecules are instead broken down to smaller fragments. Therefore, molecules which would give different molecular ions but which contain the same molecular groups will give the same m/z signal in such a mass spectrum, and will be "grouped together". If looking at an "ambient organic" mass spectrum, therefore the complexity of the mass spectrum will be reduced, simply because there might be thousands of different molecules (which in the extreme case all would give a different molecular ion mass spectral signal in the soft ionization), which however all are broken down to a limited number of "typical" fragment ions. This strongly reduces the complexity to deal with when measuring ambient organic aerosol, since "groups" of compound types can be regarded rather than individual compounds. Of course therefore no or only limited information on individual compounds can be gained with this method; for this, soft ionization methods (rather than thermal desorption / electron impact ionization) are needed, which preserve the molecular ion.

We tried to clarify this in the introduction:

"With these systems, a strong reduction in complexity of ambient organic aerosol mass spectra (which consist of a large number of different organic molecules) is achieved compared to "soft" ionisation techniques, …" (P3 L9-11),

and also in some other places as already discussed in the replies to comments above.

Line 10-13: The Kürten et al. 2007 paper states in its abstract that the instrument at that point was ready for field deployment. I am not convinced that field deployment was impossible without the modifications to the instrument described in this paper. This discrepancy needs to be resolved. Either the instrument was ready for field deployment in the 2007 paper or it wasn't. As the IT-MS was not calibrated in the field it still may not be considered "field deployable" until it can quantify aerosol components in units of mass concentration.

The Kürten et al. paper in its abstract only states that the IT-AMS "can be used as a field instrument due to its compact size". Of course in principle, it would have been possible to deploy it in the field in

that state. However, removing it from the laboratory and setting it up in the field would have been a major effort and not been possible within the typical time scale for the setup of a measurement campaign. The modifications made in order to make the instrument more reliable and less sensitive to small changes (e.g., the modifications on the electronics part, or the modifications to the helium inlet), and other modifications like the automatic control of the shutter to enable for beam open/closed measurements were indispensable in order to bring the instrument to the field with reasonable effort and to get reliable data right from the start. So, we think the statement made in the lines the reviewer comments on ("Technical improvements enable more robust and reproducible measurements … and allowed for the instrument's first field deployment.") are valid and well-founded and do not contradict the statements made in Kürten et al.. Regarding the quantification, the instrument indeed is capable of providing aerosol concentrations in units of mass concentration, see the replies to several other comments above and below.

Line 16 "and potentially quantify" this term needs to be rephrased as either a technique can quantify or it can't.

That is of course true, however the "potentially" here refers to the fact that at this stage we do not yet know whether it will be possible to truly quantify the relative fractions of sugars and carboxylic acids in more complex mixtures, since for this much more calibration work would be needed and also potential cross-sensitivities need to be determined and accounted for. So, while in principle it is possible, as we show in our manuscript, the method is not yet "finished", which we want to imply by the word "potentially". We tried to clarify this (P3, L19-21):

"which could provide a means to distinguish between carboxylic acids and sugars in organic aerosol, and with more extensive calibration potentially also to quantify their relative fractions in more complex mixtures"

Section 2: Instrumental development

- It is difficult to tell whether further advances after Fachinger 2012 were made or whether the all of the advances over the Kürten et al. 2007 paper were described by the Fachinger 2012 thesis.
- Also again since Fachinger 2012 was not peer reviewed the reader should not be referred to this thesis multiple times for clarification because the thesis is not peer reviewed and it is not in English, the language of AMT.

As described in more detail in response to several comments below, we included further technical details in order not to have to rely on the references to Fachinger (2012) any more (see revised Sect. 2). Beyond that, no further technical modifications than those also presented in the Fachinger (2012) thesis are described here. We clarified this (P4, L15-16):

"We only describe the most important changes in detail here; these as well as some other, minor modifications are also described in (Fachinger, 2012)."

We now refer to the Fachinger, 2012 thesis only at this single location in the discussion in Sect. 2.

Line 24: Since the aerodynamic lenses are referenced to Fachinger 2012, were they the same lenses used in the Kürten et al. 2007 paper? If so, please reference the Kürten paper instead of Fachinger, if not please describe the differences over the lenses in the Kürten paper.

We removed the reference to Fachinger (2012) (P3, L30), leaving only the reference to Liu et al, 1995.

Line 28: Describe to what extent the "flash-vaporization" also causes some ionization or fragmentation before EI causes further ionization and fragmentation.

We added this to the introduction, P2 L12-14:

"Due to the thermal desorption (which already might cause some fragmentation) and additional "hard" electron impact ionisation, molecules are highly fragmented, which means a partial loss of the original molecular information"

Line 32: In reference to "high purity helium" please state the purity and vendor actually used and the vendor.

We included the statement "(6.0, Westfalen AG)" on P4, L6.

Line 1: it is stated the instrument can provide quantitative information as in Kürten et al. 2007 while the last paragraph in the introduction the instrument is only potentially quantitative.

We have the impression that this is a misunderstanding: in the introduction we are referring to the potential quantification of carboxylic acids and sugars in complex mixtures of organic aerosol via MS2, which needs to be validated by more calibration work (see comment above). The statement on P4L1, on the other hand, refers to the general capability of the IT-AMS to measure quantitatively (in MS1), which was demonstrated by Kürten et al. (2007). We hope this is clarified now by the clarification of the statement in the introduction (P3, L19-21; see also reply to the corresponding comment above).

Line 2: Define what you mean here by reproducibility of the measurements. If the measurements in Kürten et al. 2007 were not reproducible doesn't that call into question the validity of the 2007 paper? Or are you referring to lack of reproducibility as a lack precision in the measurements?

The measurements by Kürten et al. are reproducible with the same instrumental settings, the problem was rather to have the instrument in a state where the settings were reproducible (e.g., measurements were taken at a certain (known) helium pressure in the trap, but this pressure was hard to maintain with the original helium inlet). So the term "measurements" is a bit misleading here; we changed it to "settings" instead (P4 L11).

Line 5: It is stated that the instrument is now more "versatile". Wasn't the instrument always capable of the measurements made in this paper? Please explain the instrument now more versatile than it was before?

The original version of the IT-AMS would have been capable of performing the measurements from the implemented hardware's point of view, but e.g. MSn measurements with n>2 had not been implemented in the previous software version, so this is one major aspect in which the modifications

made the instrument more "versatile" in practice. Also the implementation of the semi-automatic tuning in the software (which only was possible after a hardware modification) is a major step forward in practicability (and therefore also versatility) of the instrument. We now stress the need for these software changes more in the manuscript (P5 L25-26):

"The original software was very rudimentary and did not contain several important features needed for a routine deployment of the instrument."

Line 8-10: "We only describe the most important changes in detail here; other modifications are only briefly summarized and their details can be found in (Fachinger, 2012)." Since Fachinger is not peer reviewed and not written in English the details mentioned have not been scientifically reviewed and are inaccessible to all those who are not bilingual in German and English. Please either remove most of this section or provide the details here so that they can now be peer-reviewed in English.

We provided additional details in various locations, as described in response to several comments below; see revised version of Sect. 2.

Line 11-15: In the ion trap, instead of the ceramic washers used by Kürten et al. (2007) as spacers between the ring and end capelectrode, now ruby spheres (diameter 6 mm ± 0.635 μm) sitting in precise countersinks (1.3 mm ± 10 μm) are used for a more defined mounting (insert B). This allows for a more reproducible assembly of the electrodes, and consequently more reproducible voltage settings" To state an improvement the original conditions must also be stated. Please quantify the reproducibility in assembling the electrodes before and after the improvement were made. Furthermore, ion optics are electrically isolated from each other so that the ring and endcap can voltages can be set independently, how then could imprecision in electrode spacing result in a lack of "reproducible voltage settings". Please provide quantitative proof, or else only state the change without stating that it made an improvement.

The more reproducible voltage settings are due to the fact that disassembly and re-assembly of the electrodes in the old setup led to slight changes in the geometry, which causes a change in needed voltage settings for a certain operation. These changes in geometry are strongly minimized by the new, more precise method of assembly using the ruby spheres as spacers. Therefore there is no single measure to quantify the reproducibility of the assembly (since the re-tuning involves a whole set of different voltages, not a single one), but it becomes apparent in the reproducibility of the voltage settings: with the new setup, voltages have not to be re-tuned after each re-assembly of the electrodes while this was the case with the original setup. We can therefore not give a single quantitative measure of the reproducibility of the voltage settings before and after, but the reproducibility of the assembly in the old and new setups can be estimated from the tolerances of the various building elements, which we now included also for the old setup in a more detailed description (P4, L19-28):

"In the ion trap, originally ceramic washers of 2.87 mm ± 25 μm thickness were used by Kürten et al. (2007) as spacers between the ring and end cap electrode, and the electrodes were held by four threaded bars insulated by ceramic shells with a play of 0.5 mm each. Due to these rather large allowances for tolerance, the electrodes could not be assembled reproducibly enough to maintain the geometry (i.e., without rotation or tilting of the electrodes), and after each re-assembly of the electrodes the voltage settings for the various operations (e.g., trapping, scanning, resonant excitation) had to be re-tuned. To avoid this, now four ruby spheres (diameter 6 mm ± 0.635 μm)

sitting in precise countersinks (1.3 mm ± 10 µm) are used instead of the ceramic washers for a more defined mounting (insert B). This allows for a more reproducible assembly of the electrodes (i.e., invariant geometry), and consequently more reproducible voltage settings without needing to re-tune after each re-assembly of the ion trap electrodes."

Line 15-19: Please state quantitatively to what extent helium flow changed in the original setup over the course of several days. Again quantitative evidence is given for the updated system but not for the original system.

We added this information (P4, L32-33):

"over the course of four days, a relative standard deviation of 2 % was found at an average pressure of $2 \cdot 10^{-5}$ hPa measured outside the trap"

Line 18: State the orifice diameter of the critical orifice.

Done (P4, L34)

Line 19: Please state the manufacturer and flow range of the pressure-controlled mass flow controller mentioned.

We added the information requested by the reviewer (P5, L1):

"Bronkhorst High-Tech B.V., EL-PRESS P-502-C and F-004AC with a specified flow rate of ≤0.7 L min$^{-1}$"

line 22: The ion source need not be pulsed if a gate electrode between the ion source and trap is used. Here the ions in the source are continuously produced and the gate electrode voltage is lowered in order to fill the trap, and raised during ion manipulation and scanout to prevent additional ions from entering. Was this type of operation used and found deficient for some reason?

Yes, this kind of gating was also tested. However, for the IT-AMS setup, this blockage gating was not efficient enough in order to prevent all ions from entering the ion trap during the manipulation and scanning phase, which caused a stronger "background" noise in the mass spectra. Therefore, the modified method of gating the electrons entering the ion source was developed instead. We have omitted the description of this unsuccessful attempt from the manuscript in order to avoid unnecessary complexity.

Line 25: Was the filament emission current monitored? If so please quantify the instability.

The change in filament emission current was too fast to be monitored by the data acquisition software, so unfortunately no quantitative numbers can be provided.

Line 28: Better reproducibility is mentioned? State exactly what is more reproducible and how it is measured.

In order to avoid a lengthy discussion on this side-topic, we omitted the reference to better reproducibility on P5, L9.

Line 30-34: Again I suggest that since the specifics are mentioned in Fachinger 2012 that this section describing minor improvements either be expanded on in order to peer review these claims in English. Some of the points like improving the instrument housing, and improving the electrical and

communication connections seem like more trivial modifications and can be omitted unless they can be tied back to quantitative instrument improvements.

The mentioned points were anything but trivial for the performance of the instrument, but, in accordance with the reviewer's comment, we removed the information on the modifications on the electronic parts of the instruments. (P5 L19-23)

Page 5:

Line 1: Please state the version of LabVIEW used to write the program.

We added this information (v. 8.5) to the manuscript at the given location.

Line 1-6 – again since Fachinger 2012 is referenced here the nontrivial details referred to need to be provided here.

We added a more detailed description on the semi-automatic tuning and removed the reference to Fachinger, 2012. The remaining description should be understandable from the context without this reference. The paragraph now reads (P5, L24-33):

"The IT-AMS is controlled via a program written in LabVIEW (v.8.5, National Instruments), which is also utilised for data acquisition. The original software was very rudimentary and did not contain several important features needed for a routine deployment of the instrument. Therefore this software was extended and now includes the option for a semi-automatic tuning of operation parameters, i.e., the instrument is programmed by a user-adaptable text file to automatically scan the various (five for MS, nine for $MS^2$) parameters of interest and to save the results, which then can be inspected to find the optimal set of tuning parameters. The software now also allows for much more flexibility in the measurement types and their operating conditions (MS, $MS^2$, $MS^{n>2}$ ($n \leq 5$), mass range extension), programming long series of measurements, and the control of the shutter to enable automatic switching between open / closed measurements, as described above. Furthermore, all instrumental settings and parameters are now saved along with the mass spectra after each measurement cycle."

Section 3: laboratory and field measurements

Line 8: please include the stated purity and manufacturer of each chemical used. This could be done in a supplemental table or added to the existing table.

We added this information to Table 1.

Line 12: Was the Tof-AMS run in parallel with the IT-AMS so that they were both analyzing the same sample from the nebulizer? Consider making a schematic diagram of how the instruments were connected together with the nebulizer and/or with the particle counter.

The experiments were performed consecutively with the same setup, but either the IT-AMS or the ToF-AMS sampling from the nebulizer. We clarified this in P6 L2:

"In the laboratory studies performed separately with the IT-AMS and the ToF-AMS, respectively, …"

With this clarification we hope the setup is clear from the text without needing to refer the reader to an additional figure.

Line 17: Please mention to what extent each instrument was measuring the same air mass since the measurements seemed to be taken 5 meters apart and temporally (IT-MS 30s time resolution, Tof-AMS 60s time resolution). Could some of the variability in the instruments mentioned later really be due to the instruments measuring slightly different air masses either spatially or temporally?

Since at this measurement location (Mt. Kleiner Feldberg, which is a measurement location not influenced by locally emitted aerosol) only regional "background" aerosol is measured, but no locally emitted aerosol, it can be expected that both instruments sample the same type of aerosol even though their inlets are ~5m apart: in this small distance, no variation in "regional" background aerosol is expected (of course this would be different if local sources were nearby, which however was not the case). Since background aerosol is not expected to change drastically on the time scale of one minute, we do not expect the slightly different temporal resolutions of the measurements with the two instruments to have any influence on the results, and since longer averaging times of 10min and 1h are discussed, such potential differences should be averaging out anyway. We added a sentence on P6, L15-17 to discuss this:

"Since no local sources were close to the measurement site, only regional background aerosol was measured, which can be expected to be homogeneously distributed on this spatial scale."

Line 19: How can the instruments be sampling "in parallel" if they were sampling from two separate inlets? Were the inlets to each instrument both sampling from a common manifold? Or do you just mean that each instrument was sampling at the same time but from two different (although close) locations.

Yes, as is stated in this sentence: "Both instruments were sampling in parallel through two separate inlets" (P6 L14), we mean that both instruments were sampling at the same time through two separate inlets, which however were so close to another (~5m distance, P6 L15) that it can be assumed that both measured the same air masses and therefore the same aerosol type, since only regionally distributed, i.e. well-mixed aerosol and no locally emitted aerosol was probed at this measurement location (see reply to the comment above and the associated revision in the manuscript).

Also Sobanski et al 2016 doesn't mention the ToF-AMS or the IT-AMS and only refers to instruments in a molile laboratory (MoLa) for measuring aerosol parameters, thus the sentence should be rewritten to so that Sobanski only references the field campaign and not that "continuous measurements of ambient aerosol using the IT-AMS and a high resolution Tof-AMS were continuously performed" during the campaign.

We agree that this can read ambiguously, and therefore reworded to (P6, L12-14):

"Continuous measurements of ambient aerosol using the IT-AMS and a high resolution ToF-AMS were concurrently performed on the Mt. Kleiner Feldberg (Central Germany) from 29 August to 09 September 2011, within the context of a larger measurement campaign (Sobanski et al., 2016)."

Line 27 – Kürten et al 2007 states the mass range was up to 200 m/z without using the mass range extension. Did the mass range decrease over time or did the modifications stated previously lower the mass range?

The difference in the mass range reported by Kürten et al. (2007) and for our measurements is due to aging of the RF-generator, which does not (any more) provide the full voltage range it originally was specified for. With the (re-calibrated) actual voltage range the RF generator provides now, the theoretically and actually accessible mass range without mass range extension is now ~120. This is an issue of the RF generator, but is not due to the technical modifications.

Line 27-28: "ions of m/z of interest were isolated (typically within a range of ± 5 m/z) by broad band excitation using a filtered noise field" Figure 4 show isolation was outside of the ± 5 m/z range and looks more like -10 to +5 m/z in that particular instance. Please revise either your isolation range, or use a different mass spectrum in figure 4.

We changed this to (P6 L25):

"typically within a range of ± 5 *m/z*, but sometimes up to ± ~15 *m/z*"

Line 31: Please state the versions of IGOR Pro and MATLAB used.

We added the versions of IGOR and MATLAB, as requested (P6, L30)

Line 3: Please include information on how the "background effects" were performed in the methods section including how the particle free air was generated.

We included the information requested by the reviewer (P7, L1-2):

"(obtained by inserting a high efficiency particulate arrestance filter in the sampling line)."

The background effects were corrected for by adjusting the fragmentation pattern table accordingly; we clarified this in the manuscript (P6, L32 to P7, L1):

"…mass concentrations were determined using the fragmentation pattern table […], which was adjusted to correct for background effects using routinely performed measurements of particle-free air…"

Line 3: Do you have a quantitative way to state or argue that ionization efficiency in the TOF-AMS didn't change during transport of the instrument to the field location, or over the course of the campaign? Did you do any field checks of the ionization efficiency? Please add additional information on how the TOF-AMS was calibrated in Section 2.

The transport of the ToF-AMS to the measurement location should not have any influence on the ionization efficiency: the instrument was located within MoLa (designed for mobile measurements – including such of the ToF-AMS - while driving; see Drewnick et al. (2012)) where it is frequently continuously operated also during transport without any signs of changing ionization efficiency, so we do not expect any change in instrumental characteristics which could influence ionization efficiency also in this case. In the past, for example over the course of nine months, in several calibrations we observed a variation of ionization efficiency (corrected for airbeam) in the order of

~5%, which is well within the assumed uncertainty of ionization efficiency of 10%. These measurements over the course of several months include repeated measurements of ionization efficiency before and after MoLa campaigns, during which the instrument was located for several weeks in the driving vehicle and doing measurements, and no influence on ionization efficiency was observed. Therefore, we are very confident that the ionization efficiency used for the ToF-AMS was correct within the assumed uncertainty of the IE of ~10%. This is further supported by the comparison of the ToF-AMS data to measurements of co-located instruments, see reply to comment below.

As requested by the reviewer, we added some additional information on how the ToF-AMS was calibrated (P7, L2-3):

"Ionisation efficiency of the ToF-AMS was determined prior to the campaign applying the established method (Canagaratna et al., 2007) using dried $NH_4NO_3$ particles of known mobility diameter (400 nm)."

Line 4-5: Again it is vague which "co-located measurements" were taken and how "good agreement" was determined since Fachinger 2012 is not peer reviewed. Please expand this section.

We expanded this section as requested by the reviewer and now briefly include the results from the intercomparison (P7, L5-9):

"Comparison of the 1 min averaged time series of $PM_1$ calculated by summing all ToF-AMS measured species plus independently measured black carbon (using a Multi-Angle Absorption Photometer MAAP, model 5012, Thermo Scientific) with measurements of total $PM_1$ (using an Environmental Dust Monitor EDM 180, Grimm Aerosol Technik GmbH & Co. KG) gave a correlation of Pearson's $R^2$ = 0.91 and a slope of 1.11, i.e. good agreement within the uncertainty of the ToF-AMS of ~30 %."

Line 5-10: Please explain more fully (or give a reference to) how detector intensity (in units of voltage or current) is used to calculate an ion rate in units of Hz.

The channeltron in combination with a pulse discriminator, a preamplifier and a data acquisition card, as described in Kürten et al. (2007), directly measures the number of events (i.e., number of ions that were detected), so does directly provide the ion count. To obtain the ion rate, the ion count was divided by the length of the associated trapping time. We clarified this in the manuscript (P6 L28-29):

"Measured IT-AMS mass spectral signals were converted to ion rates (number of measured ions divided by the length of the trapping phase)…"

Line 16: averaging only minimizes the statistical uncertainty if the relative instrumental drift between the two instruments is negligible compared to the variation in the measurements. What evidence does the authors have in order to quantify instrument drift? It is stated that 1h average mass spectra "typically" have the same features, but do the atypical results skew the 11day average?

We did not observe any "atypical" mass spectra, but since of course we only inspected a limited number of 1h average mass spectra, we cannot exclude that there are some, and therefore qualified that statement by the word "typically". Mass spectra recorded at very low mass concentrations are more noisy and therefore do not necessarily show exactly the same patterns as mass spectra

recorded at higher concentrations on the 1h time scale, but exactly this kind of noise is averaged out by averaging over a longer period of time. When averaging over the time period of low and high mass concentrations separately (i.e., averaging out the larger contribution of noise at low mass concentrations), both show no significant differences in fragmentation patterns. We also inspected the scatter plot (IT-AMS versus ToF-AMS) for the three different species, color-coded by time of measurement, but did not observe any temporal trend in their correlation, which validates the assumption that no temporal drift in the response of the two instruments relative to each other was occurring which could bias the average over the whole time period.

Line 18: Could the difference in the instrumental response below $m/z$ 30 be due to the gating of the electrons in the IE region of the IT-AMS or different voltage settings between either the IE region and the ion trap or ToF region? Furthermore could the difference be due to decreased trapping efficiency of low $m/z$ ions.

This might be an additional influence, so we added it to the discussion (P7 L22-24):

"Apart from the potential influence of lower ion transmission or lower trapping efficiencies for low $m/z$ ions in the IT-AMS, this is probably mostly due to the strong influence of charge-transfer reactions in the ion trap during the trapping phase in this $m/z$ range…"

The original discussion regarding the influence of charge-transfer-reactions (P7 L23-25) still holds, as this concerns changes in the mass spectral patterns which are unlikely to be explained by the influences mentioned by the reviewer.

Line 23: Please quantify "much smaller"

We added the information "by more than 99 %".

Line 28 Define what it means that "this plateau disappears". For example, in the plateau disappears then does it then increase like the IT-AMS data?

We added the additional explanation "(i.e., the mass spectral pattern in this $m/z$ range becomes more similar to that of the ToF-AMS)" (P8, L1). We do not understand what is meant with the second part of the comment.

Line 3: Where these calibrations done and accounted for?

Since it was not necessary for the results presented here, we did not apply any corrections for ion transmission.

Line 4: The term "is comparible" should be quantified.

We added the information on the Pearson's $R^2$ of the mass spectral correlation (P8 L9-10):

"(Pearson's $R^2$ of 0.78; if $m/z$ 44 – which is influenced by charge-transfer reactions inside the trap, see above – is disregarded, $R^2 = 0.90$)"

Line 6: "is transferable" : the measurements may be transferrable but since there is still a large variation in the data, transferring the data would cause large uncertainties in the IT-AMS results, and

would require that a ToF-AMS always be co-located with a IT-AMS. This requirement is a major deficiency of the IT-AMS for field campaigns and should be addressed in the manuscript.

We are not sure whether we understand this comment correctly. Of course the IT-AMS measurements come with some uncertainty (as do ToF-AMS measurements). We do not see why this would mean that it is a requirement for the IT-AMS to be always co-located with a ToF-AMS during field campaigns. While it is true that the detection limit of the IT-AMS is higher than that of the ToF-AMS, this does not mean the IT-AMS results are non-quantitative without co-located ToF-AMS measurements. While the ToF-AMS gives better temporal resolution due to higher signal to noise ratio, the IT-AMS gives more in-depth information, while still providing the general, quantitative information on the main aerosol components also measured by the ToF-AMS. So, we do not understand why it should be a requirement to have a co-located ToF-AMS, and therefore also do not see the "deficiency" the reviewer mentions.

Line 9: is "m/z 48 to 64" supposed to read, "m/z 48 and 64" since only m/z 48 and 64 are designated at sulphates in the upper part of figure 2?

We mean the signal ratio (m/z 48 / m/z 64) here, i.e., "m/z 48 to m/z 64" within the context of the sentence is correct. We reworded the sentence to make this clearer (P8 L15):

"The signal ratio $m/z$ 48 to $m/z$ 64…"

Line 15-16: "No calibration measurements were performed for the IT-AMS." This is a major deficiency of the manuscript as this means the IT-AMS mass concentrations must be tied to the ToF-AMS which was also not calibrated in the field. The authors must explain why neither instrument was not calibrated in the field and intercompared using the same aerosol generation source.

While the ToF-AMS was not calibrated on-site, it was properly calibrated directly prior to the measurement campaign in the laboratory, and that calibration can be reasonably assumed to be valid also at the measurement site (see reply to comment above). The ToF-AMS calibration is also validated by the comparison to co-located measurements by other instruments (see reply to comment above). Therefore, we are very confident that the mass concentrations provided by the ToF-AMS are accurate within its instrumental uncertainty. Since the intercomparison of both instruments over the 11-day long measurement campaign did not show any temporal trends in the relationship between the measurements of the two instruments, any time interval out of this field campaign could be used as calibration – similar to a calibration with laboratory-generated aerosol. For the remaining time the IT-AMS would provide independent quantitative measurements without the need for a co-located ToF-AMS (see also reply to several comments above).

Line 20: Please comment on the fact that the 1h R2 values for nitrate are lower than that of the 10min data.

This apparent decrease (0.68 for 10min, 0.65 for 1h) reflects the uncertainty of the reported $R^2$ values. We tested the change of $R^2$ when using different averaging times (1 to 60 min), and found that after an initial rise in $R^2$ with longer averaging time, the correlation did not improve from a certain averaging time on any more. While for sulphate and organics $R^2$ still improved slightly at larger averaging times than 10min, for nitrate, the "plateau" of approximately constant $R^2$ was already reached at 10min averaging time, so the slight difference in $R^2$ for 10min and 60min averaging time simply reflects the associated variability / uncertainty.

Since this does not seem like a major issue, we omitted this discussion from the manuscript.

Line 24: It should be noted in the manuscript that Drewnick et al. 2009 were calculating the detection limits for a TOF-AMS and not a IT-MS

The method described in Drewnick et al. (2009) is based on theoretical considerations which are mass analyzer independent and therefore also hold for the IT-AMS. Indeed, the method was tested (by Drewnick et al., 2009) not solely for the ToF-AMS, but also for the Q-AMS. Since the method described by Drewnick et al. is based solely on theoretical considerations without using any assumptions specific to a ToF-AMS, we think the wording "were calculated following the method by Drewnick et al. (2009)" is sufficient.

Line 25-30: Using the "linear" relationships in figure 3 to calculate mass concentrations for the IT-AMS will produce large uncertainties in the limit of detection for the IT-AMS (especially for sulphate). The uncertainty in the detection limits must be calculated and reported. Also if the R2 value for the 1h measurements of nitrate are larger than the 10min data, how can the limit of detection for the 1hr data be lower than the 10 minute data for nitrate. Furthermore, if it was claimed in the previous paragraph that the IT-AMS signal for sulfate is lower than that of nitrate, and that sulfate has a lower signal to noise ratio compared to nitrate, how can sulphate and nitrate have the same detection limit? This discrepancy needs to be explained in the manuscript.

Of course every calibration is accompanied by a certain uncertainty. This is not fundamentally different for a calibration using the ToF-AMS than for a calibration using size-selected particles measured in parallel with a CPC. We can estimate the uncertainty of the derived IT-AMS mass concentrations from the combination of the following contributing uncertainties:

1.) Uncertainty due to the scatter of datapoints of ToF-AMS and IT-AMS. This is visible in the scatter of the datapoints around the linear correlation line, and accounted for by the uncertainty of the slope derived in the fit. This uncertainty of the slope was found to be <6 % for the three investigated species (typically even <4%); we therefore assume an uncertainty of 6 % as a conservative estimate.

2.) Uncertainty of the ToF-AMS used as a reference. This uncertainty (30 % total) is derived from three major uncertainties: Uncertainty of the CE (~25 %), uncertainty of the RIE (~10 %), and uncertainty of the IE (~10 %) ($\sqrt{10\%^2 + 10\%^2 + 25\%^2} \approx 30\%$). This uncertainty therefore already accounts for uncertainty of the CE which also needs to be applied for the IT-AMS; the uncertainty of RIE and IE would also have been accounted for when calibrating the IT-AMS using e.g. size-selected particles and a CPC as reference instrument.

From these two uncertainties, the total uncertainty of the IT-AMS can be estimated to $\sqrt{6\%^2 + 30\%^2} = 30.6\%$, i.e., the total uncertainty is about the same as the uncertainty derived for the ToF-AMS (~30%).

Note that the overall uncertainty would be approximately the same when calibrating the instrument e.g. using size-selected particles and a CPC: in that case, RIE, IE and CE uncertainty would need to be accounted for as for the ToF-AMS, resulting in an overall uncertainty of ~30% as well.

Therefore, we can conclude that the total uncertainty of the mass concentrations of the IT-AMS is almost the same as the total uncertainty of ToF-AMS measurements. Moreover, the fact that the instrument was calibrated using the ToF-AMS and not e.g. using a CPC and size-selected particles

does not result in a significantly higher uncertainty, due to the fact that the uncertainties accounted for in the total uncertainty of the ToF-AMS would still need to be accounted for.

We included the uncertainty estimate of the derived mass concentrations in the manuscript (P9 L12-14), and also included the absolute uncertainties of the LODs derived for the different species (P9 L27-29):

"The uncertainty of the derived mass concentrations of the IT-AMS can be estimated to 30 % (which includes the uncertainty due to ionisation efficiency, relative ionisation efficiency, and collection efficiency), the same as usually estimated for ToF-AMS measurements."

"Detection limits for 10 min averages are $(3.7 \pm 1.1)$ µg m$^{-3}$ for organics, $(1.3 \pm 0.4)$ µg m$^{-3}$ for nitrate, and $(1.3 \pm 0.4)$ µg m$^{-3}$ for sulphate (for 1 h averages: $(1.4 \pm 0.4)$ µg m$^{-3}$ for organics, $(0.5 \pm 0.2)$ µg m$^{-3}$ for nitrate, $(0.7 \pm 0.2)$ µg m$^{-3}$ for sulphate)."

As discussed in response to the comment above, the difference in $R^2$ value for the 10min and 1h averaged time series of nitrate is not significant and solely reflects the uncertainty of the $R^2$ value.

The discrepancy the reviewer mentions regarding the comparable LOD of sulfate and nitrate despite the fact that sulfate has a lower signal to noise ratio can be explained by the uncertainties of the LODs, as discussed above. Since we now include the uncertainty of the LODs in the discussion in the manuscript, we hope this apparent discrepancy is resolved for the reader.

Section 4.2 and following

These sections are the highlight of the paper in my opinion and quite well reasoned.

Section 4.3.2

The authors make a convincing argument that MS2 studies can differentiate sugars and carboxylic acids for fragment ion isomers. However, it seems more straight forward to differentiate these species based on their molecular ions in the MS1 spectrum as all of the species in table 1 have different molar masses. The manuscript could be also be improved by differentiating molecular ion isomers by looking for unique fragments of each isomer, or differences in the ratio of fragments for isomers that do not have unique fragments.

We agree that in principle the differentiation of the compounds used for this study would be possible by their molecular ions, however this is not a feasible option for atmospheric measurements with this kind of instrument. Otherwise, every ToF-AMS should be able to differentiate these compounds in atmospheric aerosol, which is not the case. The problem with this type of instrument is that due to the strong fragmentation (thermal desorption / EI), the molecular ion signal, if present at all, is very small. Therefore, while ion traps coupled to "softer" ionization sources are used to differentiate molecular ions of different isomeric structure by their fragmentation pattern, this is not a feasible option for the IT-AMS (or ToF-AMS) with its thermal desorption / EI ion source. However, the approach we use here is similar to the one suggested by the reviewer, only that isomers not of the molecular ion, but of certain fragment ions are differentiated by their MS2 spectra in order to draw conclusions on the structure of the parent molecule.

In the ambient atmosphere, typically a large variety of different molecules will contribute to the same m/z, so while it of course is possible to differentiate pure compounds in the lab by their mass

spectra (MS1), this is not an option for atmospheric applications (or laboratory applications where mixtures of organic compounds are studied, like in smog chamber experiments). Furthermore, the very fact that all sugars of different original structure will generate the same fragment ions at m/z 60 and 73, and all carboxylic acids as well will give the same fragment ions at these m/z, which however can be distinguished from those of sugars due to their different isomeric structure, is a major advantage over other aerosol mass spectrometers, since in atmospheric applications mixtures of various different sugars / carboxylic acids can be expected. Since they all will produce the same fragment ions, which can be apportioned by the method described here to sugar / carboxylic acid, this yields information which would not be possible to obtain with a ToF-AMS.

So, rather than providing information on the abundance of single compounds, the IT-AMS provides information on a whole compound class at once, which, depending on the atmospheric application, is a major advantage over current instrumentation.

Section 5

line 11-12: It is stated that that the detection limit "were found to be sufficiently low to allow for MS2 studies on atmospheric particles under favorable ambient conditions (0.6 micrograms/m3)". The 0.6 figure seems to come not from either ambient measurements or from MS2 measurements. The 0.6 figure seems to come from the MS1 measurement of tryptophan under laboratory conditions. I would suggest either referring to the limits of detection for organics, nitrates, and/or sulphates from the field campaign, or the MS2 detection limits from the tryptophan measurements which were stated as being 7 micrograms/m3

Thank you for catching this! This indeed was a typographical error, and should have read "7 µg m$^{-3}$" instead. We have corrected this in the revised manuscript.

Figure 1: Please add the pressure into the ion trap region, and show in which sections turbopumps or other pumps are located. In subpart "A" show that He comes from a compressed gas cylinder and what the flow rate of He is into the ion trap. For "B" it looks like these figures are taken directly from Fachinger 2012 and should be referenced as such.

The pumps were omitted on purpose to avoid cluttering the figure with too much detail, but we added them back in.

We added a symbol for the gas cylinder, as requested by the reviewer, and also added the flow rate of the helium to the ion trap as well as the pressure in the ion trap region.

We added the reference to the Fachinger 2012 thesis in the figure caption, as suggested by the reviewer.

See Figure 1 in the revised manuscript.

Figure 2: Please make it clear in the text or caption how the relative signal was calculated. Are these ratios of raw detector output or ratios of ion rates? Also it is confusing after reading the caption if the top graph is calculated from the relative signals from the bottom graph. If this is so then the bottom and top graphs should be switched. I suggest changing the bottom graph's y-axis label to "signals relative to m/z 28. Also from the caption it is unclear as to how the normalization occurred. Was the

IT-MS data normalized to the m/z 28 value from the IT-AMS, and the ToF-AMS data normalized to the m/z 28 from the Tof-AMS, or was data from both the IT-AMS, and Tof-AMS normalized to the IT-AMS m/z 28 signal. Also there seems to be some data that is missing such as m/z 13 and 15 that show up in the bottom graph. Below m/z 20 some of the ToF-AMS data is larger than the IT-AMS which obscures the IT-AMS data completely. M/z 21 for the Tof-AMS is not existent in the bottom graph the m/z 21 is shown the top graph. M/z 88 shows a very small signal for the Tof-AMS compared to the IT-AMS signal, but in the top graph m/z 88 has a similar relative signal to m/z 87, and 89. Unless I am reading the graph incorrectly, I am not sure how this discrepancy could occur. Finally, since data below m/z 30 is not useful in differentiating organics, sulphates, or nitrates you may consider omitting this data and only briefly referring to it in the text of the manuscript.

We swapped the top and the bottom graph, as suggested by the reviewer. We also changed the y-axis label to "relative signal intensity (relative to m/z 28)" for clarification, and clarified in the figure caption that both mass spectra were normalized to their respective signal at m/z 28. The ratios are calculated from the final corrected data (difference mass spectra in ion rates, normalized to m/z 28), which we now also state in the figure caption:

"Comparison of average difference mass spectra measured with the IT-AMS and the ToF-AMS during 11-day long ambient measurements. Shown are the average mass spectra normalised (after conversion to ion rates) to their respective mass spectral signal at $m/z$ 28 (upper panel) and the ratio (IT-AMS to ToF-AMS) of these relative signal intensities, colour-coded for the dominant species at the respective $m/z$ (lower panel)."

Some data in the graph showing the mass spectra are not visible due to the range of the y-axis scale (logarithmic scaling beginning at $10^{-5}$). Values which are positive but smaller than $10^{-5}$ and not visible in the mass spectra due to that scaling are still displayed in the panel showing the ratio of IT-AMS to ToF-AMS. Such is the case for m/z 21 the reviewer mentions. Changing the scaling of the y-axis would result in less visible differences between the "large" peaks, while the information gained on the small peaks is not important for the presented discussion, so we decided to show a clearer presentation of the more important signals instead. We clarified this in the figure caption:

"Note the logarithmic scaling of the y-axes and that the upper panel's y-axis only starts at $10^{-5}$ (i.e., ion signals smaller than that are not shown)."

The m/z missing in the panel which shows the ratios IT-AMS to ToF-AMS (m/z 13 and 15 the reviewer mentions) are due to negative values in one of the mass spectra (therefore giving a negative ratio, which is not displayed in the logarithmic scaling), which can occur due to the calculation of difference mass spectra. As also the reviewer states, these m/z (<30) are not very important for the presented discussion, so we think it is not detrimental to the figure as a whole if some of these m/z are not visible in one of the graphs. Still, we think it is useful to include also m/z <30 in the figure since this can be used to illustrate the discussion in Sect. 4.1 and makes it easier to follow.

The observation the reviewer made regarding m/z 87, 88 and 89 is explained by the logarithmic scaling of the shown mass spectra.

Figure 3: For the sulphate graph please align the 0 measurement for the IT-AMS with the 0 measurement of the ToF-AMS like it is for the nitrate and organics graphs. It seems as though some of the data for the IT-AMS might give negative ion rates and thus under the detection limit for the

instrument. Please also include a horizontal dotted line on the IT-AMS graph indicating the calculated limit of detection of the instrument. It looks like the data for nitrates after September 8th might also be below the detection limit, so this should be discussed in the text. Finally, local date and time, should be designated in the caption vs UTC so that the reader knows the time zone and if local time includes daylight savings time.

We thank the reviewer for catching the differences in y-axis scaling of sulphate, which we now corrected to align for both instruments. Additionally, we now scaled all IT-AMS time series in a way that both the measured ion rate, and the mass concentration calculated from this ion rate can be obtained for the IT-AMS from the y-axes. We also made the other requested changes: we added the IT-AMS detection limits, and changed the x-axis label to "local date and time (UTC+2)". We also changed the figure caption accordingly to reflect these changes. See Fig. 3 in the revised manuscript. As pointed out by the reviewer, some of the 1h averages are below the LOD; however, averaging the whole time period (as done for the average mass spectrum) lowers LOD such that these data still are meaningful to be included in the long-term average.

Figure 4. The vertical dotted line in graph a) should be defined in the caption. It looks like the dotted line is used to differentiate a region to the left which uses the left y-axis, and a region to the right that uses a different y-axis. Please include only a graph of the isolation step before graph b).

We changed the figure caption according to the reviewer's comment and included an additional panel (b) which shows the mass spectrum after the isolation step, as requested by the reviewer. See Fig. 4 in the revised manuscript.

Specific comments pertaining to format:

Line 7: "The currently most widely…" suggested rephrase to "Currently, the most widely"

done

line 10: delete "here" and change "is" to "are" in the sentence "Since here a large number of different molecules is analysed simultaneously"

done

Line 13: Suggest changing "strongly fragmented" to "highly fragmented"

done

Line 18: "high resolution" to "higher resolution" as high resolution mass spectrometry often refers to Penning trap instruments such as ICR or Orbitrap which have resolutions orders of magnitude greater than TOF. It would also be good to state here what the resolution of IT-AMS is from the Kürten et al. 2007 paper.

Thank you! Good point regarding the ambiguity of "high" resolution. We changed this accordingly and also added the resolution of the IT-AMS, as requested (P4 L7-8):

"Mass spectral resolution depends on the settings (Kürten et al., 2007) and was ~400 for the measurements described here."

Line 25: consider removing "some" in "While some information on elemental composition".

done

line 15: Please rephrase this sentence so that it doesn't end in "with."

done

line 5: remove the word "regular" in "regular ToF-AMS

done

page 12 line 30: replace "on" with "in"

done

**Referee #2:**

The manuscript by Fachinger et al. describes technical improvements applied to a previously developed ion trap aerosol mass spectrometer (IT-AMS). The IT-AMS measures the chemical composition of aerosol particles by means of flash vaporization followed by electron impact ionization and ion analysis with an IT mass spectrometer. While the mass spectrometer and ion detection unit were developed in-house, the vacuum chamber with its aerodynamic lens inlet and the vaporization/ionization unit are identical to the Aerodyne AMS vacuum chamber. The most commonly used commercially available aerosol mass spectrometers nowadays use time of flight (ToF) mass spectrometers, which e.g. allow to derive information on the origin of the organic fraction of the aerosol constituents due to the high mass resolving power of a ToF. However, an IT has the capability of performing so-called MSn studies, which can help to distinguish between fragments with different isomeric structures. Therefore, the IT-AMS has, in principle, an important advantage over other commercially available aerosol mass spectrometers.

The capabilities of the IT-AMS are demonstrated by lab experiments, where generated organic particles are analyzed by MSn studies. The different fragmentation patterns show clear differences between different compound classes. Furthermore, measurements during a field campaign indicate very similar results for the IT-AMS and a ToF-AMS for the nitrate, organic and sulfate fraction of ambient aerosols.

Regarding the description of the modifications, I agree with referee #2 that these need to be explained in much greater detail. Just referring to a German PhD thesis is not sufficient, especially since the chosen journal (AMT) is actually well-suited for a description of technical details that improve a measurement technique.

However, overall the paper is well-written and should be published in AMT after addressing the comments listed in the following as well as the requested improvement on describing technical details.

We thank the reviewer for his/her valuable comments. The reasoning behind keeping the technical descriptions rather short and referring the reader to the PhD thesis instead was to keep the paper short and easier to read. However, we agree that the paper should be able to stand by itself without needing to refer to the PhD thesis, and therefore have added additional technical details in Sect. 2 (see the revised version of the manuscript).
* * *
Specific comments:

Page 1, line 19: change „was demonstrated" to "is demonstrated"

done

Page 2, line 3: the paper by Schramm et al. (2009) should also be cited in this respect, especially since the paper makes uses of a similar ion trap as used in the present study

We thank the reviewer for the suggestion, and added a reference to the Schramm et al. paper in the discussion on P3, L5.

Page 4, line 12: please mention how many ruby spheres were used

We specified to "four ruby spheres".

Page 4, line 27: please explain what "modified" exactly means

We added a more detailed description on the difference between the original and the modified filament. We also added some more discussion on the deflection plate, as per the next comment.

"In the original filament, the emission of the electrons (defined by the filament current) and the voltage of the filament's deflection plate were electronically coupled in such a way that emitted electrons always were repelled by the deflection plate and accelerated away from the filament and towards the ion cage. In the modified filament the deflection plate is electronically decoupled from the filament, such that electrons are emitted continuously, but the voltage of the deflection plate can be set independently and switched from negative to positive sign. By this means, electrons emitted by the filament are now either deflected or absorbed by this deflection plate (insert C), depending on whether they are needed in the ion source or not. This controlled absorption of the electrons (instead of only repelling them from the ion cage) allows for a more defined gating of the electrons and avoids the potential build-up of space charges." (P5, L10-18)

Page 4, line 29: the reason for using the deflection plate and its functionality should be discussed

See reply to previous comment.

Page 5, line 2/3: please explain how the semi-automatic tuning of operation parameters works

We added an explanation on the semi-automatic tuning to the manuscript (P5 L26-29):

"Therefore this software was extended and now includes the option for a semi-automatic tuning of operation parameters, i.e., the instrument is programmed by a user-adaptable text file to automatically scan the various (five for MS, nine for MS[2]) parameters of interest and to save the results, which then can be inspected to find the optimal set of tuning parameters."

Page 5, line 26: up to what m/z can the mass range be extended? Up to what m/z was the set-up tested?

Kürten et al. (2007) demonstrated the mass range extension up to m/z 1000. In the current setup, it was tested up to ~m/z 300. Since this is already discussed by Kürten et al. (2007), we omitted it from the present manuscript.

Page 6, line 9/10: what about other compounds like water, ammonium and chloride? If the usable mass range starts at 30 amu, it means that the important compound class ammonium cannot be measured as in the standard AMS, please discuss

This is correct, with this setup ammonium cannot be determined. Non-refractory chloride (in atmospheric applications typically $NH_4Cl$) in principle should be detectable (m/z 35 and 36), but under typical ambient conditions is very low in concentration (during the measurement discussed here the average Chl concentration measured with the AMS was 0.04 µg/m3, i.e., below the expected detection limit of the IT-AMS). We added a short discussion on this in P 9, L14-18:

"Note that with the IT-AMS, unlike the ToF-AMS, ammonium mass concentration cannot be determined due to artefacts in the $m/z$ range <$m/z$ 30, as described above. Another species typically reported from ToF-AMS measurements, non-refractory chloride, in principle should be possible to detect with the IT-AMS (dominant mass spectral lines at $m/z$ 35 and 36), but has not been observed during this measurement due to very low mass concentrations (campaign average of 0.04 µg m[-3] found with the ToF-AMS)."

Page 7, line 1: should there be a "≥" sign instead of ">"?

We are discussing signals ≥m/z 30, but since m/z 30 is defined in this context as being dominated by nitrate, the signals assigned to "organics" (which we are referring to here) are at m/z >30.

Page 7, line 20-23: this cannot be the only explanation as the signals are well above the LOD (1.3 / 0.7 µg/m3), any other ideas?

We added a potential additional explanation to this discussion (P8, L30-31):

"Also the fact that the observed range of mass concentrations for sulphate was smaller than for organics and nitrate might have added to the less tight correlation for this species."

Page 8, line 26: what is the fraction of the doubly charged ions? where are they coming from (from the ion source or from reactions inside the ion trap)?

We did not perform dedicated measurements in order to quantify the fraction of doubly charged ions, but from our measurements we can provide a lower limit of ~10 %. We added this information ("to at least ~10 %", P10 L28) to the manuscript.

We do not know for sure where these ions are coming from, but since EI as "hard" ionization technique is known to produce singly as well as doubly charged ions, we think that they are originating from the ion source rather than from reactions inside the ion trap.

Page 9, line 24: please explain better how the value of the ion recovery is exactly determined

We added a short explanation on how ion recovery is defined here:

"i.e., the total signal of all fragment ions detected in $MS^2$ divided by the concurrent loss in signal of the parent ion" (P11, L26-27)

Page 10, line 27: please add "and pinonic acid" after „… (pink color)"

Done

Figure 1: (i) the photographs in panel b) are too small, (ii) the functionality of the deflection plate shown in panel c) should be explained in more detail in the text

We enlarged the photographs (see Figure 1 in the revised manuscript), as requested by the reviewer. The functionality of the deflection plate is now explained in more detail in Sect. 2 (P5, L10-19, see reply to comment above).

Figure 2: is this the final data after correcting for the backgrounds, etc.? it says the IT-AMS has a lower signal to noise, but why are the IT-AMS signals larger by a factor of 10? if this is the case then the sensitivity could be significantly improved by reducing the noise; what is causing the high noise?

These are the finalised mass spectra corrected for duty cycles and background effects (i.e., difference mass spectra of open and closed). We added this information to the figure caption. – Regarding the signal intensities: note that both mass spectra are normalized to their signal intensity at m/z 28. Due to the fact that the IT-AMS mass spectrum shows lower relative contribution of m/z 28 (due to reactions in the ion trap, see also discussion in the manuscript), the relative signal intensities of the other ions are higher than in the ToF-AMS mass spectrum, which explains the apparently higher (but only relative to m/z 28!) signal intensity in the shown mass spectrum. We changed the figure caption to make this normalization to m/z 28 clearer for the reader:

"Comparison of average difference mass spectra measured with the IT-AMS and the ToF-AMS during 11-day long ambient measurements. Shown are the average mass spectra normalised (after conversion to ion rates) to their respective mass spectral signal at *m/z* 28 (upper panel) and the ratio (IT-AMS to ToF-AMS) of these relative signal intensities, colour-coded for the dominant species at the respective *m/z* (lower panel)."

Figure 3: sulfate: memory effects? heater temp. the same?

The heater temperature for both instruments was ~600 °C. If there were any memory effects regarding sulphate, such effects should already have been accounted for by the calculation of difference mass spectra, i.e. changing background signal is continuously corrected for. We are not

aware of other, unaccounted for memory effects which could systematically detriment the measurement of ammonium sulphate present in atmospheric aerosol particles.

Figure 6 and Figure 7: explain what "fraction" means as the fragments do not seem to add up to 100%

For the MS2 results, "fraction" refers to the intensity of the fragment ion relative to the most intense fragment ion, as explained in the figure caption. For the ToF-AMS results, "fraction" means the relative contribution of the various fragment ions to the UMR m/z. The missing contributions the reviewer commented on are caused by other fragment ions which contribute to the UMR m/z to a small extent, but are not displayed in these figures for clarity. We clarified this in the figure captions:

"On the left, the relative contributions of $[C_3H_3O]^+$ and $[C_4H_7]^+$ to $m/z$ 55 and of $[C_3H_5O]^+$ and $[C_4H_9]^+$ to $m/z$ 57 are given (from ToF-AMS measurements; the difference to 100 % is due to one or several other ions contributing little to the respective $m/z$)."

[revised manuscript text omitted]

---

## Author Response (AR2)

I thank the authors for their extensive revisions and comments in the first round of peer review.

Please address Figure 4b,c in regards to more than m/z130 undergoing colission induced dissociation in the MS2 experiment. I noticed in particular that m/z129 and m/z131 are also removed during collision induced dissociation which may give provide an additional source of m/z 83. Right now it assumed that m/z 83 and 84 arise only from fragmentation of m/z 130, but it seems that at least m/z 83 could also arise from m/z 129. The authors should address this point in the text of the manuscript and/or in the figure caption. Furthermore a m/z range should be given over which ions undergo collision induced dissociation (looks right now like + or - 1 m/z).

We added this to the discussion in Sect. 4.2 (P10, L7-8):

"It has to be noted that CID in these experiments is found to potentially affect ions in a range of ± 1 around the $m/z$ of interest, i.e., in the case of glutathione at least parts of the signal at $m/z$ 83 could also originate from CID of $m/z$ 129."

Finally some of the y-axis numbers seem to overlap a bit when more than one graph is stacked on on top of the other (for example in fig. 4 b,c). Please rescale or adjust the axis numbers to prevent overlap.

We thank the reviewer for the hint and have changed Figs. 3, 4 and 5 accordingly.

After these points are address I believe the article will be in a publishable state.

[revised manuscript text omitted]